# LOW-DIMENSION-TO-HIGH-DIMENSION GENERALIZATION AND ITS IMPLICATIONS FOR LENGTH GENERALIZATION

## ABSTRACT

Low-Dimension-to-High-Dimension (LDHD) generalization is a special case of Out-of-Distribution (OOD) generalization, where the training data are restricted to a low-dimensional subspace of the high-dimensional testing space. Assuming that each instance is generated from a latent variable and the dimension of the latent variable reflects the problem scale, the inherent scaling challenge in length generalization can be captured by the LDHD generalization in the latent space. We theoretically demonstrate that LDHD generalization is generally unattainable without exploiting prior knowledge to provide appropriate inductive bias. Specifically, we explore LDHD generalization in Boolean functions. We verify that different architectures trained with (S)GD converge to *min-degree interpolators w.r.t. different linearly independent sets*. LDHD generalization is achievable if and only if the target function coincides with this inductive bias. Applying the insights from LDHD generalization to length generalization, we explain the effectiveness of CoT as changing the structure latent space to enable better LDHD generalization. We also propose a principle for position embedding design to handle both the inherent LDHD generalization and the nuisances such as the data format. Following the principle, we propose a novel position embedding called RPE-Square that remedies the RPE for dealing with the data format nuisance.

## 1 INTRODUCTION

The field of learning to reason has gained significant popularity in the machine learning community due to its impressive performance in reasoning tasks such as natural language processing (OpenAI, 2023a;b), mathematics (Frieder et al., 2023; Jelassi et al., 2023), coding (Zhang et al., 2022a), symbolic logic (Abbe et al., 2023; Garcez et al., 2022), and planning (Zhao et al., 2023; Valmeekam et al., 2023). Reasoning problems inherently require generalization beyond the training distribution, a challenge known as Out-of-Distribution (OOD) generalization (Krueger et al., 2021; Ye et al., 2021). One of the most significant challenges in OOD generalization is scaling, also known as *length generalization* (Anil et al., 2022; Zhang et al., 2022b), where models trained on small-scale instances must generalize to large-scale instances. Length generalization is crucial because the size of sample spaces often increases exponentially with the complexity of reasoning problems, leading to intractable sample complexity and computational costs for models that do not achieve length generalization.

Numerous works have investigated the length generalization problem in various tasks and proposed various techniques that help scaling in practice, including modifications to model architectures (Shaw et al., 2018; Jelassi et al., 2023; Kazemnejad et al., 2024), transformations in data formats (Lee et al., 2023; Zhou et al., 2023), prompt engineering for Large Language Models (LLMs) Wei et al. (2022); Feng et al. (2024). While some techniques work uniformly well across a wide class of problems, many are fragile and even ad-hoc, applicable only to specific problems with certain formats (Zhou et al., 2024). This can be largely attributed to the mismatch between the inherent problem scale and the formal scale in the input domain (e.g., the length of the input string in language models). We further elaborate the statement in Example 1.

**Example 1.** *Consider the addition learning problem in the language domain, where the input is a string. For an $N$-digit-plus-$N$-digit (written as $N$-addition) addition of two numbers $x$ and $y$, where $x = x_{N-1} \ldots x_0, y = y_{N-1} \ldots y_0, x_i, y_i \in \{0, \ldots, 9\}$ ($x_{N-1} > 0$ or $y_{N-1} > 0$), consider two*

*formats of the instance: the Aligned and Reverse Format (ARF), where the instance is represented as "$\mathtt{x}_0 \ldots \mathtt{x}_{n-1} + \mathtt{y}_0 \ldots \mathtt{y}_{n-1} =$"; and the Unaligned and Reverse Format (URF), where the instance is represented as "$\mathtt{x}_0 \ldots \mathtt{x}_{\mathtt{n}_x-1} + \mathtt{y}_0 \ldots \mathtt{y}_{\mathtt{n}_y-1} =$", $n_c = \arg\max_i\{c_i \neq 0\}, c \in \{x, y\}$. The formal scale in the language domain is the length of the string. However, the length of the input strings in neither format faithfully reflects the inherent problem scale: in ARF, the length of the input strings is always invariant; in URF, a shorter string may correspond to a larger scale than a longer string (e.g., $s_1 =$"$1 + 1234 =$" is 4-addition, $s_2 =$"$123 + 123 =$" is 3-addition; however, $|s_1| < |s_2|$). Furthermore, we find in experiments that Relative Position Embedding (RPE) improves the length generalization performance of Transformers for ARF, while it provides little benefit for the length generalization of Transformers for URF.*

To develop robust and transferable methods for scalable models, it requires a formulation that captures the inherent scaling challenge of length generalization and is invariant to the nuisances such as data format. To do so, we first notice that in the length generalization of a wide range of problems, the instances can be seen as generated from some latent space whose dimension reflects the scale.

**Example 2** (Addition). *Let $\Sigma = \{0, \ldots, 9\} \times \{0, \ldots, 9\}$. The instance of $n$-addition $x_{n-1} \ldots x_0 + y_{n-1} \ldots y_0$ can be represented by the state vector $h_n = [(x_0, y_0), \ldots, (x_{n-1}, y_{n-1})]$. The $k$-th dimension of $h_n$ corresponds to the $(k-1)$-th digits of the two addenda (i.e., $x_{k-1}$ and $y_{k-1}$). The expansion of the state vector in the dimension corresponds to the increase in the addendum digits. Therefore, the length generalization from $N_0$-addition to $N$-addition can be seen as a generalization from the low-dimensional latent space $\Sigma^{N_0}$ to the high-dimensional latent space $\Sigma^N$.*

To catch the above intuition, we make the following assumption about the underlying data generation.

**Assumption 1** (Data Generation in Length Generalization). The data generation process of an instance of scale $n$ with the concept $c$ is as follows:

1. A latent variable $h$ is sampled from a subspace $\Sigma^n$ of dimension $n$;

2. The label $y$ is determined by the concept $c$ and the latent variable $h$, i.e., $y = c(h)$;

3. The hidden variable $h$ is transformed to an input sequence by the data format mapping $\phi$.

According to Assumption 1, the inherent scaling challenge can be captured by Low-Dimension-to-High-Dimension (LDHD) generalization, a special scenario of OOD generalization with structural distributional shifts. In an LDHD generalization problem, a model is trained with samples from a low-dimensional subspace while evaluated with samples from a higher-dimensional superspace. See Figure 2 in Appendix B for an intuitive illustration of LDHD generalization, compared to in-distribution and typical OOD generalization. To further illustrate, we can consider that the training samples are from the space $\Sigma^{N_0}$ and the testing samples are from $\Sigma^N$, where $\Sigma$ is a domain and the numbers $N_0, N$ satisfy $N_0 < N$. With slight abuse of notation, we also use $\Sigma^n$ to denote the embedding space of the subspace $\Sigma$ into a space $\Sigma^{\bar{N}}$ for some sufficiently large $\bar{N} \geq N > N_0$ (we use $\bar{N} = N$ below if there is no special statement).

The main challenge of LDHD generalization is that the testing space has extra dimensions compared to the training space. As a result, the testing space contains instances with orthogonal components to the training space. The training samples can reveal no information about how these components contribute to the results. For instance, in Example 2, learning with $N_0$-addition solely does not tell how $(N_0 + 1)$-th digits to $N$-th digits contribute to the result unless we have the prior knowledge that the addition of each digit shares the same process. We formally this challenge as No-Free-Lunch Theorem of LDHD Generalization in Section 3.

The No-Free-Lunch Theorem (Wolpert & Macready, 1997; Wolpert, 2002) of LDHD Generalization necessitates the use of prior knowledge in the learning process in order to achieve LDHD generalization. In practice, the prior knowledge is usually incorporated via the inductive bias of the learning algorithms and the models. Abbe et al. (2023) claimed that models with (S)GD converge to min-degree interpolators. However, the minimal degree-profile bias neither holds uniformly for all model architectures nor is sufficient for achieving LDHD generalization. To develop LDHD generalizable models, we investigate different architectures whose inductive bias with (S)GD can potentially go beyond the min-degree interpolators. While random feature models are shown to be min-degree interpolators, we prove that random feature models with projections, where inputs are transformed

by projections before fed to the random feature models, converge to *min-degree interpolators under different linearly independent sets*. We further consider Position-Only Linear Attentions with Advice (PLAA), i.e., linear attentions whose attention scores are determined only by positions, with additional hints about the scales of the instances. This model can be seen as a simplified abstraction of decoder-only transformers with a special focus on positional relations that are considered crucial for length generalization. We demonstrate the failure of PLAA models equipped with Absolute Position Embeddings (APEs) in general cases and show the PLAA models with Relative Position Embeddings (RPEs) are min-degree interpolators w.r.t. some linearly independent sets. These results show that for models biased towards min-degree interpolators w.r.t. some linearly independent sets, LDHD generalization can be achieved if and only if the target mapping is the min-degree interpolator w.r.t. the same set.

The LDHD generalization perspective further provides insights on how Chain-of-Thought (CoT) (Wei et al., 2022) can help length generalization. We show that CoT can be seen as a change of the hidden space, where each dimension of the latent space is augmented with an additional middle state. This transformation could facilitate LDHD generalization in the latent space for some problems. Another implication is a principle of position embedding design for length generalization with Transformers: we need to handle both the inherent LDHD generalization and the nuisances such as the data format in the design of the position embeddings. Following this principle, we propose a novel position embedding named RPE-Square. The RPE-Square enhances the RPE with the ability to handle certain data format nuisances, evidently improving the length generalization of the URF addition over the Transformer with the RPE.

The main contributions of the paper are summarized as follows:

- We propose to study LDHD generalization, which captures the inherent scaling challenge of a wide range of length generalization problems. LDHD generalization provides a formulation under which we can perform theoretical analysis for general length generalization problems.

- We study the inductive bias of different model architectures under (S)GD, which can be exploited to incorporate prior knowledge to achieve LDHD generalization. In the context of Boolean functions, we show that different architectures under (S)GD can have inductive bias other than min-degree interpolators, which enables LDHD generalization for a wider class of problems.

**Notations.** We use $[n]$ to represent the set of numbers $\{1, \ldots, N\}$. We denote the set of all functions from the set $\mathcal{X}$ to the set $\mathcal{Y}$ as $\mathcal{F}_{\mathcal{X}, \mathcal{Y}}$. We define $\mathrm{Proj}(x, V)$ as the coordinate of the projection of $x$ onto the space spanned by $V$ with the basis $V = [v_1, \ldots, v_r]$, i.e., $[\mathrm{Proj}(x, V)]_i = \langle x, v_i \rangle$ for all $i = 1, \ldots, r$. We use $A^*$ to denote the Kleene closure of the set $A$, i.e., $A^* = \bigcup_{k=0}^{\infty} A^k$. We use $\deg(p)$ to denote the degree of the polynomial $p$. We represent the set of $N \times N$ upper triangular matrix as $\mathcal{U}_N$.

## 2 RELATED WORK

**Length generalization in reasoning problems.** Length generalization is a key challenge in learning to reason, typically interpreted as the ability to learn with small-scale instances of a task and generalize to unseen large instances of the same task (Anil et al., 2022; Zhou et al., 2023). Various reasoning tasks are considered to investigate length generalization, including arithmetic (Jelassi et al., 2023; Feng et al., 2024), Boolean logic (Abbe et al., 2023; d'Ascoli et al., 2023), symbolic reasoning (Zhang et al., 2022b), etc. Despite the rich literature on length generalization on specific reason tasks, few works have considered challenges and overconditions of length generalization for general problems. The existing works that analyze general length generalization mainly focus on the change of the input sequence length (Xiao & Liu, 2023; Ahuja & Mansouri, 2024), which does not precisely capture the difficulties of length generalization. Our work proposes to consider LDHD generalization, which characterizes the inherent challenges of length generalization in many tasks.

**Inductive Bias of Model Architectures and Algorithms.** In the situation of learning with over-paratermization or incomplete information, proper inductive bias is essential to select the true model from the hypothesis (Neyshabur, 2017; Bartlett et al., 2021). One of the most studied scenarios is the inductive bias of different model architectures under (S)GD. Previous research shows (S)GD combined with different model architectures lead to different effects of implicit regularization, such as

different norms of the learnable parameters Gunasekar et al. (2017); Bartlett et al. (2021) and different complexity characterizations of the models (Razin & Cohen, 2020; Razin et al., 2021; Abbe et al., 2023). We specially mention Abbe et al. (2023), which proposes that models trained with (S)GD are biased towards min-degree profile interpolators in the context of Boolean functions, which do not achieve general LDHD generalization. Our results show that the min-degree profile bias does not hold for all models. We further show that models with different architecture can converge to min-degree profile interpolators under different linearly independent sets when trained with (S)GD. This partially explains how model architectures can affect length generalization in reasoning problems.

# 3 LOW-DIMENSION-TO-HIGH-DIMENSION GENERALIZATION

**Definition 1** (Low-Dimension-to-High-Dimension Generalization). Suppose that $\mathcal{X}$ is a sample space and $\mathcal{X}_1, \mathcal{X}_2$ are two subspaces such that $\mathcal{X}_1 \subset \mathcal{X}_2 \subset \mathcal{X}$ and $\dim(\mathcal{X}_1) < \dim(\mathcal{X}_2)$. Consider a concept class $\mathcal{C} \subset \mathcal{F}_{\mathcal{X},\mathcal{Y}}$, two distributions $\mathcal{D}_1, \mathcal{D}_2$ where $\text{supp}(\mathcal{D}_1) = \mathcal{X}_1$ and $\text{supp}(\mathcal{D}_2) = \mathcal{X}_2$, and a learning algorithm $\mathcal{A} : (\mathcal{X} \times \mathcal{Y})^* \mapsto \mathcal{F}_{\mathcal{X},\mathcal{Y}}$. We say *low-dimension-to-high-dimension generalization* of the concept class $\mathcal{C}$ from $\mathcal{D}_1$ to $\mathcal{D}_2$ is achieved by the algorithm $\mathcal{A}$ with $m$ samples and $\epsilon$ error if

$$\mathbb{E}_{X^m \sim \mathcal{D}_1^m, X_{m+1} \sim \mathcal{D}_2} \left[ \ell \left( \hat{f}_{X^m,c}(X_{m+1}), c(X_{m+1}) \right) \right] \leq \epsilon, \tag{1}$$

where $\hat{f}_{X^m,c}$ is the function learned by the algorithm $\mathcal{A}$ from the training samples $X^m$ labeled by the concept $c$ and $\ell : \mathcal{Y} \times \mathcal{Y} \mapsto \mathbb{R}$ is the loss function.

Definition 1 extends the Independent Identical Distribution (IID) assumption in PAC learning theory and considers a special shift between the training data and the testing data. This shift is particularly challenging because the testing space is of strictly higher dimension than the training space. Generally, it is impossible to fully capture the structure of the training space from the testing data as the training data reveal no information on how components in the orthogonal subspace contribute to the output. Therefore, there is no algorithm that can always guarantee to learn the concept from the training data. Theorem 1 formally states the nonexistence of universal algorithms for LDHD generalization.

**Theorem 1** (No-Free-Lunch Theorem of LDHD Generalization). *Suppose that the two sets $\mathcal{X}$ and $\mathcal{Y}$ are finite. For some $N > N_0$, consider two subsets $\mathcal{X}_{N_0}$, $\mathcal{X}_N$ of $\mathcal{X}$ such that $\mathcal{X}_{N_0} \subsetneq \mathcal{X}_N \subseteq \mathcal{X}$ and $\dim(\mathcal{X}_{N_0}) = N_0 < N = \dim(\mathcal{X}_N)$. Let $c_1, c_2 \in \mathcal{F}(:= \mathcal{F}_{\mathcal{X}_N,\mathcal{Y}})$ be two concepts such that $c_1(x) = c_2(x)$ for all $x \in \mathcal{X}_{N_0}$. For any $c \in \mathcal{F}$ and $\mathcal{X}' \subseteq \mathcal{X}$, define $\mathcal{F}/(c \mid \mathcal{X}') := \{f \in \mathcal{F} \mid f(x) = c(x) \text{ for all } x \in \mathcal{X}'\}$. Let $\ell : \mathcal{Y} \times \mathcal{Y} \mapsto \mathbb{R}$ be the loss function. For any distribution $\mathcal{D}(\mathcal{X}_N)$ such that $\text{supp}(\mathcal{D}(\mathcal{X}_N)) = \mathcal{X}_N$:*

$$\sum_{f \in \mathcal{F}/\left(c_1 \mid \mathcal{X}_{N_0}\right)} \mathbb{E}_{x \sim \mathcal{D}(\mathcal{X}_N)} \left[ \ell\left(c_1(x), f(x)\right) \right] = \sum_{f \in \mathcal{F}/\left(c_2 \mid \mathcal{X}_{N_0}\right)} \mathbb{E}_{x \sim \mathcal{D}(\mathcal{X}_N)} \left[ \ell\left(c_2(x), f(x)\right) \right].$$

Theorem 1 necessitates the consideration of structural assumptions on the concept class such that a learning algorithm could identify the target concept from the hypothesis with the imperfect information provided by the low-dimensional training data. For example, the concept class of linear classifiers with fixed weight vector on $\mathcal{X} = \mathbb{R}^d$, i.e., $\mathcal{C} = \{\text{sgn}(w_0^\mathsf{T} x + b) \mid w_0 \in \mathbb{R}^d, b \in \mathbb{R}\}$ with the $d'$-dimensional training sample space $\mathcal{X}_1 = \mathbb{R}^{d'} \times \{0\}^{d-d'}$ and the $d$-dimensional testing sample space where $d' < d$. For any concept $c \in \mathcal{C}$, a learning algorithm could not identify the true concept $c$ solely from the training data from $\mathcal{X}_1$ and the hypothesis of all linear classifiers. However, if a linear algorithm exploits the structure of the concept class that the weight vector always equals $w_0$, it can compute the target bias from the training data in $\mathcal{X}_1$ and thus identify the target concept $c$.

We further investigate how model structures can affect LDHD generalization. We show in Section 4 that different models trained with (S)GD can be seen as min-degree interpolators under different functional bases in the context of Boolean functions, which is a joint inductive bias of the model structures and (S)GD. This insight suggests a principle for model design to achieve LDHD generalization: ensure the concept class is "low-degree" under the linearly independent set induced by the model structure.

Many generalization challenges in reasoning problems can be attributed to LDHD generalization inherently. Length generalization is representative of these challenges and of significance in reasoning

problems. Intuitively, length generalization means that a model trained with small-scale instances of a reasoning problem can perform well on large-scale instances of the problem. To formalize the intuition, we consider a typical instance being generated from a hidden state variable $h$ that represents the core of the instance and is transformed to the model input by a data format mapping. The hidden state space is of the form $\Sigma^*$ for some domain $\Sigma$. The dimension $n$ of the subspace from which the hidden state variable is sampled, $n = \arg\max_k \{h \in \Sigma^k, h \notin \Sigma^{k+1}\}$, reflects the increase in the scale of the problem. Besides, we define the concept class on the hidden space, depicting that the change of the data format does not change the intrinsic logic of the concepts. Assumption 1 describes the pipeline of the data generation in length generalization.

## 4 MAIN RESULTS

We show theoretically how different models succeed or fail to achieve LDHD generalization as the effect of the inductive bias of the architectures under (S)GD. We focus on Boolean functions. More specifically, in the context of Boolean functions, we have $\Sigma = \{\pm 1\}$, $\mathcal{X} = \Sigma^N$, $\mathcal{X}_n = \Sigma^n \times \{1\}^{N-n}$ for $n = 1, \ldots, N$. The set of all Boolean functions potentially considered is $\mathcal{F} = \mathcal{F}_{\mathcal{X}, \mathbb{R}}$. We consider LDHD generalization from $N_0$ to $N$ for some $N_0 < N$. Define $I(f)$ as the minimal set $I$ of indices that the function $f$ can be represented as a function of $x_I$, i.e., $I(f) := \arg\min_{I \subset [N]} |I|$ such that $f(x) = \tilde{f}(x_I)$ for some function $\tilde{f}$ and all $x \in \mathcal{X}$. We say a function $f$ is $k$-sparse if $|I(f)| \leq k$.

Before presenting the theoretical results, we introduce two concepts *degree profile w.r.t. linearly independent set* and *min-degree interpolator w.r.t. linearly independent set*, which extend the concept *degree profile* and the concept *min-degree interpolator*, respectively. We use the two concepts to characterize the inductive bias of different model architectures under (S)GD.

**Definition 2** (Degree Profile w.r.t. Linearly Independent Set $\mathcal{B}$)**.** Suppose that $\mathcal{B} = \{b_1, \ldots, b_R\}$ is a linearly independent set of functions in $\mathcal{F}$ and $D = \max_{b \in \mathcal{B}} \deg(b)$. Let $f \in \mathcal{F}$ be a function in the subspace spanned by $\mathcal{B}$, i.e., $f = \sum_{i=1}^{R} \hat{f}_{\mathcal{B}}(b_i) b_i$ for some $\hat{f}_{\mathcal{B}}(b_i) \in \mathbb{R}, i = 1, \ldots, R$. The *degree profile* of the function w.r.t. $\mathcal{B}$, denoted by $\mathrm{DegP}_{\mathcal{B}}(f)$, is a $(D+1)$-tuple where $D_i = \sum_{b \in \mathcal{B}, \deg(b) = D+1-i} \hat{f}_{\mathcal{B}}(b)^2$ for $i = 1, \ldots, D+1$. The order of degree profiles is identical to the lexicographic order of the corresponding $D$-tuples.

**Definition 3** (Min-Degree Interpolator w.r.t. Linearly Independent Set $\mathcal{B}$)**.** Suppose that $\mathcal{B} = \{b_1, \ldots, b_R\}$ is a linearly independent set of functions in $\mathcal{F}$. Let $\mathcal{X}'$ be a subset of the sample space $\mathcal{X} = \{\pm 1\}^N$. Denote the set of all interpolators on $\mathcal{X}'$ for the concept $c$ by $\mathcal{G}_{\mathcal{X}', c}$, i.e., $\mathcal{G}_{\mathcal{X}', c} = \{g \in \mathcal{F} \mid g(x) = c(x) \text{ for all } x \in \mathcal{X}'\}$. A function $g$ is called the min-degree interpolator w.r.t. $\mathcal{B}$ on $\mathcal{X}_0$ for the concept $c$ if $g \in \mathcal{G}_{\mathcal{X}', c}$ and $\mathrm{DegP}(g) \leq \mathrm{DegP}(g')$ for all $g' \in \mathcal{G}_{\mathcal{X}', c}$.

### 4.1 RANDOM FEATURE MODEL WITH PROJECTION

We first consider the random feature model (RFM) and a class of its variants, i.e., Random Feature Models with Projection (RFMP; see Definition 4). RFM is widely employed as approximations of practical neural network models in theoretical studies. By comparing the inductive biases introduced by RFM and RFMP under various projections, we demonstrate the importance of incorporating prior knowledge to achieve LDHD generalization and this prior knowledge can be effectively embedded through model design.

**Definition 4** (Random Feature Model with Projection)**.** Suppose that $V = [v_1, \ldots, v_r] \in \mathbb{R}^{N \times r}$ satisfies $V^\mathsf{T} V = I_r$. A random feature model with projection w.r.t. $V$ is

$$f_{\mathrm{RFMP}}^{V, K}(x; a) = \frac{1}{\sqrt{K}} \sum_{k=1}^{K} a_k \sigma\left(\langle w_k, \mathrm{Proj}(x, V)\rangle + b_k\right),$$

where $K$ is the number of random features, $a = [a_1, \ldots, a_K]^\mathsf{T}$ is the learnable parameter, $w_k \sim \mathcal{N}(0, I_r/r)$, $b_k \sim \mathcal{N}(0, 1/r)$ for $k = 1, \ldots, K$, and $\sigma$ is the activation function.

The original RFM can be seen as a special instance of RFMP with $V = I_N$. Technically, we follow the strongly expressive condition (Abbe et al., 2023) for the activation function $\sigma$. Abbe et al. (2023) shows that the RFM converges to the min-degree interpolator when initialized at 0 and trained with GD. However, this is not the case for all RFMP models. We show in Theorem 2 that an RFMP model converges to a min-degree interpolator w.r.t. a linearly independent set determined by the set $V$.

**Theorem 2.** *Suppose that $V = [v_1, \ldots, v_r] \in \mathbb{R}^{N \times r}$ satisfies $V^\mathsf{T} V = I_r$. Define the set $\mathcal{B}(V)$ of independent functions as*

$$\mathcal{B}(V) = \left\{ \chi_T^V(x) \right\}_{T \subseteq [r]}, \quad \text{where } \chi_T^V(x) = \prod_{t \in T} \sum_{n=1}^{N} (v_t)_n x_n$$

*Let $a_t$ be the learnable parameter at the timestep $t$ in the training process where the learnable parameter $a$ is initialized at $a_0 = 0$ and optimized with gradient descent/gradient flow under $\ell_2$ loss on $\mathcal{X}_{N_0}$. Let $\mathcal{G}_{N_0,c,V}$ be the set of all interpolators on $\mathcal{X}_{N_0}$ for the concept $c^*(x) = c\left(\text{Proj}(x, V)\right)$ that is $O_N(1)$-sparse. Then we have*

$$f_{RFMP}^{V,K}(x; a_t) \to \arg \min_{g \in \mathcal{G}_{N_0,c,V}} \text{DegP}_{\mathcal{B}(V)}(g), \quad \text{as } K \to \infty, t \to \infty.$$

When $V = I_N$, the linearly independent set $\mathcal{B}(V)$ is the Fourier basis of the Boolean functions, and Theorem 2 implies that the RFM converges to the min-degree interpolator. From Theorem 2, we see that an RFMP model with the projection matrix $V$ can achieve LDHD generalization only if the target concept coincides with the min-degree interpolator w.r.t. $\mathcal{B}(V)$. Specially, for the RFM, we have:

**Corollary 1.** *For any $f \in \mathcal{F}$ such that $I(f) \not\subset [N_0]$, the min-degree interpolator does not achieve LDHD generalization from $\mathcal{X}_{N_0}$ to $\mathcal{X}_N$ and thus the RFM initialized at 0 and trained with GD does not achieve LDHD generalization from $\mathcal{X}_{N_0}$ to $\mathcal{X}_N$.*

Corollary 1 shows that the min-degree interpolator and thus the RFM model can only achieve LDHD generalization for a very restricted set of functions that are only dependent on $x_{[N_0]}$. To achieve LDHD generalization with RFMP models requires prior knowledge of the concept class to design a projection to align the concepts with the min-degree interpolator w.r.t. the projected Fourier basis. Example 3 illustrates how LDHD generalization is possible for the target function with dependence beyond $x_{[N_0]}$ by choosing a proper projection.

**Example 3.** *Consider the target function $f(x) = 4x_1 + 3x_2$, $N_0 = 1$, and $N = 2$. The min-degree interpolator on $\mathcal{X}_{N_0}$ is $f_1(x) = 4x_1$, which does not achieve LDHD generalization on $\mathcal{X}_N$. In the RFMP model, if we choose*

$$V = \begin{bmatrix} 0.8 & 0.6 \\ 0.6 & -0.8 \end{bmatrix},$$

*then we have $\mathcal{B}(V) = \{1, 0.8x_1 + 0.6x_2, 0.6x_1 - 0.8x_2, (0.8x_1 + 0.6x_2)(0.6x_1 - 0.8x_2)\}$. The min-degree interpolator w.r.t. the linearly independent set $\mathcal{B}(V)$ on $\mathcal{X}_{N_0}$ is $f_2(x) = 4x_1 + 3x_2 = f(x)$, which achieves LDHD generalization on $\mathcal{X}_N$.*

### 4.2 POSITION-ONLY LINEAR ATTENTION WITH ADVICE

In this subsection, we investigate Position-Only Linear Attention with Advice (PLAA), which can be seen as a simplification of decoder-only Transformers (Definition 5), with a special focus position embeddings that are considered pivot to the length generalization of the Transformers (Shaw et al., 2018; Jelassi et al., 2023).

**Definition 5** (PLAA). Define the advice function $n : \mathcal{X} \mapsto \{0, \ldots, N\}$ such that $n(x) = \arg \max_n \{x_n = -1\}$ if there exists $k \in [N]$ such that $x_k = -1$ and $n(x) = 0$ otherwise. We additionally define $e_0 := 0$. A PLAA model is

$$f_{\text{PLAA}}(x; A) = x^\mathsf{T} A e_{n(x)},$$

where $A \in \mathcal{U}_N$ is the learnable parameter and $e_n$ denotes the vector with a 1 in the $n$-th coordinate and 0's elsewhere.

We further elaborate on the intuition behind the PLAA models. In the generation process of a decoder-only (linear) Transformer with position embeddings given input $\mathtt{s} = \mathtt{s_1} \ldots \mathtt{s_n}$, the attention is computed by the query at the position $n$ and the keys at the positions $i \leq n$. The position of the query is special, advising the length of the input and reflecting the scale of the instance ideally. The PLAA model captures this feature and introduces the notation $n(x)$ to reflect the dimension of the subspace that $x$ belongs to. To further simplify and focus on the position embeddings, we assume

that the value of each $x_i$ is identical to itself (i.e., we fix the value matrix in the attention to $I$ and thus $W_V x_i = x_i$) and the attention is only related to positions. The contribution of the interaction between the position embeddings is $[e_1, \ldots, e_n]^\mathsf{T} A_{[n],[n]} e_n = A_{[n],n}$ for some upper triangle matrix $A$. When the input is embedded to length $N$ but the query is still made at the position $n$, the output of the model is $x^\mathsf{T} A e_{n(x)}$, i.e., the output of the PLAA model. Therefore, the PLAA model is a simplification of decoder-only Transformers that enables theoretical analysis of the impact of the position embeddings on length generalization. For a more detailed elaboration on PLAA, see Appendix C.

In Definition 5, we directly parameterize the PLAA model directly with the attention matrix. In practice, however, the attention matrix is typically computed by the interaction between the position embeddings. Therefore, we consider the PLAA models with the Absolute Position Embedding (APE) and the Relative Position Embedding (RPE), respectively. See Definitions 6 and 7. Note that we consider Generalized RPE (GPRE) in Definition 7 because the RPE can be seen as a special instance of the GRPE with $\mathcal{U} = \mathcal{U}_{\text{RPE}} = \{D_1, \ldots, D_N\}$, where $D_k$ is a $k$-th upper diagonal matrix such that $(D_k)_{ij}$ is 1 if $j = i + k - 1$ and 0 otherwise. We seek a more general result applicable to all similar parameterization methods to the RPE.

**Definition 6** (PLAA with APE). A PLAA model with APE is

$$f_{\text{PLAA}}^{\text{APE}}(x; P) = x^\mathsf{T} \left( M_N^u \circ P^\mathsf{T} P \right) e_{n(x)},$$

where $M_N^u \in \mathbb{R}^{N \times N}$ is the upper triangle mask, i.e., $(M_N^u)_{ij}$ is 1 if $i \leq j$ and 0 otherwise, and $P \in \mathbb{R}^{d_P \times N}$ is the learnable parameter of the model.

**Definition 7** (PLAA with GRPE). For $\mathcal{U} = \{U_1, \ldots, U_r\}$, a PLAA moodel with GRPE is

$$f_{\text{PLAA}}^{\text{GRPE},\mathcal{U}}(x; p) = x^\mathsf{T} \left( \sum_{i=1}^{r} U_i p_i \right) e_{n(x)},$$

where $U_i \in \mathcal{U}_N, i = 1, \ldots, r$ are upper triangle matrices that satisfy $\langle U_i, U_i \rangle = 1$ for all $i = 1, \ldots, r$ and $(U_i)_{kl} (U_j)_{kl} = 0$ for all $i \neq j$ and $1 \leq k, l \leq N$, and $p = [p_1, \ldots, p_r]^\mathsf{T} \in \mathbb{R}^r$ is the learnable parameter of the model.

*Remark* 1. The condition $(U_i)_{kl} (U_j)_{kl} = 0$ for all $i \neq j$ and $1 \leq k, l \leq N$ in Definition 7 means each element in the position-only attention is characterized by at most one parameter. For example, in RPE, $A_{i,j}(i \leq j)$ is only parameterized by $p_{j-i}$. This condition naturally generalizes the intuition of RPE that characterizes some "relative" relations between two positions.

We first consider the vanilla PLAA model, the direct parameterization with the attention matrix. Theorem 3 shows that the PLAA model converges to the min-degree interpolator w.r.t. the linearly independent set $\mathcal{B}_N^{\text{PLAA}}$ that is defined in the theorem.

**Theorem 3.** *Define the set $\mathcal{B}_N^{PLAA}$ as*

$$\mathcal{B}_N^{PLAA} := \{b_{ij}^{PLAA}(x)\}_{1 \leq i \leq j \leq N},$$

*where*

$$b_{ij}^{PLAA}(x) = \begin{cases} -\frac{1-x_j}{2} \prod_{k=j+1}^{N} \frac{1+x_k}{2}, & i = j, \\ x_i \frac{1-x_j}{2} \prod_{k=j+1}^{N} \frac{1+x_k}{2}, & i < j. \end{cases}$$

*Let $A_t$ be the learnable parameter at the timestep $t$ in the training process where the learnable parameter $A$ is initialized at $A_0 = 0$ and optimized with gradient descent/gradient flow under $\ell_2$ loss on $\mathcal{X}_{N_0}$. Let $\mathcal{G}_{N_0, A^*}^{PLAA}$ be the set of all interpolators on $\mathcal{X}_{N_0}$ for the concept $c(x) = f_{PLAA}(x; A^*)$. Then we have*

$$f_{PLAA}(x; A_t) \to \arg \min_{g \in \mathcal{G}_{N_0, A^*}^{PLAA}} \text{DegP}_{\mathcal{B}_N^{PLAA}}(g), \quad as\ t \to \infty.$$

For the PLAA model with APE, Theorem 4 shows that the model converges to the interpolator whose attention matrix is of the minimal nuclear norm if the minimal-nuclear-norm-solution conjecture Assumption 2 about the implicit regularization in matrix factorization holds. While the minimal-nuclear-norm-solution conjecture has not been theoretically justified in general cases, it is proved correct under some specific conditions and supported by empirical evidence (Gunasekar et al., 2017; Arora et al., 2019).

**Assumption 2** (Minimal-Nuclear-Norm-Solution Conjecture (Gunasekar et al., 2017)). Consider the optimization

$$\min_{U \in \mathbb{R}^{n \times n}} f(U) = \| \mathcal{A} \left( UU^{\mathsf{T}} \right) - y \|_2^2,$$

where $\mathcal{A} : \mathbb{R}^{n \times n} \mapsto \mathbb{R}^m$ is a linear operator specified by $\mathcal{A}(\cdot)_i = \langle A_i, \cdot \rangle$ for symmetric linearly independent $A_i \in \mathbb{R}^{n \times n}, 1 \leq i \leq m$, and $y \in \mathbb{R}^m$. Define $X = X(U) := UU^{\mathsf{T}}$. Let $U_t$ be the parameter reached by gradient flow in the timestep $t$ and $X_t$ be $U_t U_t^{\mathsf{T}}$. Define $X_\infty(X_{\text{init}}) := \lim_{t \to \infty} X_t$ for $U_0 U_0^{\mathsf{T}} = X_{\text{init}}$. For any full rank $X_0$, if $\hat{X} = \lim_{\alpha \to 0} X_\infty(\alpha X_0)$ exists and $\mathcal{A}\left(\hat{X}\right) = y$, then $\hat{X} \in \arg\min_{\mathcal{X} \succ} \|X\|_*$ s.t. $\mathcal{A}(X) = y$.

*Remark 2. Some works (Gidel et al., 2019; Li et al., 2021) propose that rank minimization is more accurate than nuclear norm minimization under certain technical assumptions. While the nuclear norm minimization and the rank minimization are not always equivalent, both can explain the failure of APE in length generalization in certain scenarios. Furthermore, in the literature on low-rank optimization, the nuclear norm has been shown to serve as an effective surrogate for rank minimization under particular assumptions (Recht et al., 2008; Candès & Tao, 2010; Candes & Recht, 2012), making it a widely used regularization technique in practical applications. In this work, we focus on the length generalization problem and assume Assumption 2 for simplicity.*

**Theorem 4.** *Suppose that Assumption 2 holds and $d_P = N$. Let $P_t$ be the learnable parameter at the timestep $t$ in the training process where the learnable parameter $P$ is initialized at $P_0$ such that $(P_0)_{ij} \sim \mathcal{N}(0, \alpha)$ for $(i, j) \in [d_P] \times [N]$ and $\alpha$ sufficiently small, and optimized with gradient descent/gradient flow under $\ell_2$ loss on $\mathcal{X}_{N_0}$. Let $\mathcal{G}_{N_0, P^*}^{APE}$ be the set of all PLAA interpolators on $\mathcal{X}_{N_0}$ for the concept $c(x) = f_{PLAA}^{APE}(x; P^*)$. Then we have*

$$f_{PLAA}^{APE}(x; P_t) \to \arg \min_{g \in \mathcal{G}_{N_0, P^*}^{APE}} \|A_g\|_*, \quad as\ t \to \infty,$$

*where $A_g$ is the attention matrix in the PLAA model $g$.*

Theorem 4 shows that APE leads to a low-rank attention matrix in the PLAA model. Therefore, when the training data is restricted to $\mathcal{X}_{N_0}$, the rank of the attention matrix learned by the model with APE is no higher than $N_0$. This leads to length generalization failure when the target concept requires higher-rank attention to be represented. We have the following direct corollary of Theorem 4 that partially explains the limitation of APE for length generalization.

**Corollary 2.** *The PLAA with APE fails to achieve LDHD generalization for the concept $c(x) = f_{PLAA}(x; A^*)$ if $\text{rank}(A^*) > N_0$.*

Theorem 5 characterizes that the PLAA converges to the interpolator that minimizes the degree-profile w.r.t. to the linearly independent set $\mathcal{B}_{\text{PLAA}}^{\text{GRPE}, \mathcal{U}}$, which can be seen as the projections of $\mathcal{U}$ onto $\mathcal{B}_{\text{PLAA}}^{\text{GRPE}, \mathcal{U}}$. Theorem 3 is a special case of Theorem 5 for $\mathcal{U} = \{E_{ij}\}_{1 \leq i \leq j \leq N}$, where $E_{ij} \in \mathbb{R}^{N \times N}$ is the elementary matrix that has a 1 at the $(i, j)$-th position and 0 everywhere else.

**Theorem 5.** *For the $\mathcal{U} = \{U_1, \ldots, U_r\}$, define*

$$\mathcal{B}_{PLAA}^{GRPE, \mathcal{U}} := \left\{ \sum_{1 \leq i \leq j \leq N} (U_k)_{ij} b_{ij}^{PLAA}(x) \right\}_{1 \leq k \leq r}.$$

*Let $p_t$ be the learnable parameter at the timestep $t$ in the training process where the learnable parameter $p$ is initialized at $p_0 = 0$ and optimized with gradient descent/gradient flow under $\ell_2$ loss on $\mathcal{X}_{N_0}$. Let $\mathcal{G}_{N_0, p^*}^{GRPE, \mathcal{U}}$ be the set of all interpolators on $\mathcal{X}_{N_0}$ for the concept $c(x) = f_{PLAA}^{GRPE, \mathcal{U}}(x; p^*)$. Then we have*

$$f_{PLAA}^{GRPE, \mathcal{U}}(x; p_t) \to \arg \min_{g \in \mathcal{G}_{N_0, p^*}^{GRPE, \mathcal{U}}} \text{DegP}_{\mathcal{B}_{PLAA}^{GRPE, \mathcal{U}}}(g), \quad as\ t \to \infty.$$

With the inductive bias of the PLAA with GRPE, Corollary 3 states that LDHD generalization can be achieved if and only if the target concept can be represented by the elements in $\mathcal{B}_{\text{PLAA}}^{\text{GRPE}, \mathcal{U}}$ that have dependence on $\mathcal{X}_{N_0}$.

**Corollary 3.** *Consider the concept $c(x) = f_{PLAA}^{GRPE,\mathcal{U}}(x; p^*) = \sum_{k=1}^{r} c_k \sum_{1 \le i \le j \le N} (U_k)_{ij} b_{ij}^{PLAA}(x)$. Under the conditions of Theorem 5, the PLAA with GRPE achieves LDHD generalization from $\mathcal{X}_{N_0}$ to $\mathcal{X}_N$ if and only if*

$$\left\{ k \mid (U_k)_{[N_0],[N_0]} = 0 \right\} \subseteq \left\{ k \mid c_k = 0 \right\}.$$

*Remark* 3. While the projection in the RFM and the position embeddings in the PLAA may introduce a stronger inductive bias that benefits LDHD generalization, they can also reduce the expressiveness of the models. The critical point is to align the models with the concept class in such a way that we can incorporate a strong inductive bias for LDHD generalization while maintaining sufficient expressiveness for the concept class.

## 5 FURTHER IMPLICATIONS FOR LENGTH GENERALIZATION

We discuss further implications of the LDHD generalization perspective for length generalization.

### 5.1 CHAIN-OF-THOUGHT FOR LENGTH GENERALIZATION

While the Chain-of-Thought (CoT) can lead to more variety in the length of testing samples, it is widely used as an effective technique to improve the length generalization in various reasoning tasks. This seems contradictory if considered in the original space of the input sequence. However, when considering the abstraction of the data generating process from the hidden space, we can see that CoT intrinsically changes the underlying hidden space by extending each dimension with a "middle" variable and does not lead to the dimensional increase in the hidden space. For example, consider the $n$-addition without CoT and the $n$-addition with CoT. In the case without CoT, the instance $x_{n-1} \ldots x_0 + y_{n-1} \ldots y_0 = z_n \ldots z_0$ corresponds to the latent state $h_n = [(x_0, y_0), \ldots, (x_{n-1}, y_{n-1})] \in \Sigma^n$ for $\Sigma = \{0, \ldots, 9\} \times \{0, \ldots, 9\}$. In the case with CoT, one step of predicting $z_t$ in solving the instance corresponds to the latent state $h_n = [(x_0, y_0, z_0), \ldots, (x_{t-1}, y_{t-1}, z_{t-1}), (x_t, y_t, *) \ldots, (x_{n-1}, y_{n-1}, *)] \in \bar{\Sigma}^n$ for $\bar{\Sigma} = \Sigma \times \{*, 0, \ldots, 9\}$, where $*$ is a special element indicating undetermined values. CoT does not cause the LDHD generalization challenge but extends the domain $\Sigma$, which could potentially lead to a more easily learnable target concept under the model architecture.

### 5.2 POSITION EMBEDDINGS FOR LENGTH GENERALIZATION

Position embeddings are considered closely related to the length generalization in Transformers Our analysis suggests a principle to consider both the inherent LDHD generalization and the nuisances such as the data format in the design of the position embeddings. To further elaborate, consider the length generalization of the URF addition with CoT. While RPE can capture the recursive structure of the addition problem and could lead to LDHD generalization in the latent space, it fails to work for the URF addition with CoT. The failure of RPE is attributed to the nuisance of the URF.

Following the principle, we design a novel position embedding called RPE-Square to handle the nuisance of the URF. On the one hand, we keep the RPE structure for the inherent LDHD generalization. On the other hand, we deal with the unalignment by considering the distances to several special tokens (i.e., [BOS], +, and =). These considerations lead to the RPE-Square, in which we compute the relative values between the distances to the special tokens. More concretely, the RPE-Square$_{i,j}$ for the query at $j$ and the key at $i$ is

$$\sum_{1 \le k \le j, 1 \le l \le i} \frac{\exp\left((W_Q x_j)^\intercal (W_K x_l)\right)}{\sum_{1 \le l' \le j} \exp\left((W_Q x_j)^\intercal (W_K x_{l'})\right)} \frac{\exp\left((W_Q x_i)^\intercal (W_K x_k)\right)}{\sum_{1 \le k' \le i} \exp\left((W_Q x_i)^\intercal (W_K x_{k'})\right)} R_{(j-l)-(i-k)},$$

where $W_Q$ and $W_K$ are the weight matrices for the query and the key, respectively. We replace RPE$_{j-i}$ with RPE-Square$_{i,j}$ in the Transformer. The experiment shows that RPE-Square effectively improves the length generalization in the URF addition with CoT. The design of RPE-Square illustrates how the principle can lead to position embeddings for better length generalization. More details of the RPE-Square and the experiments can be found in Appendix E.

## 6 CONCLUSION AND DISCUSSION

We propose considering LDHD generalization, which captures the inherent scaling challenge of length generalization in various reasoning tasks. We introduce the No-Free-Lunch Theorem for LDHD

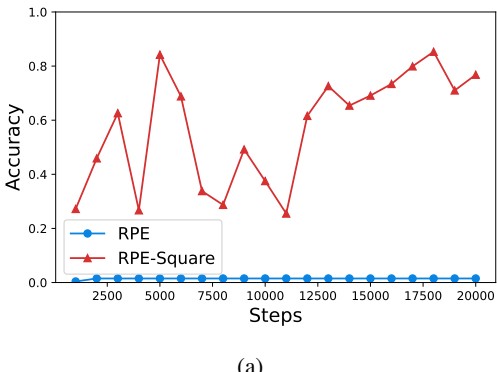 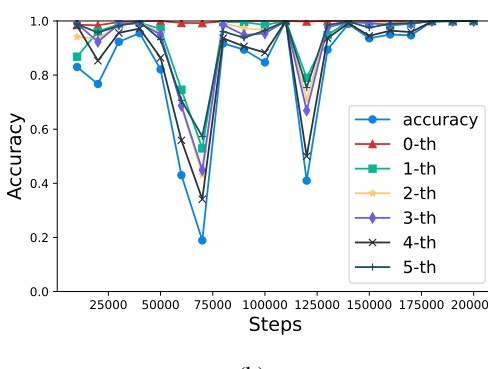

(a)                  (b)

Figure 1: Length generalization of Transformer with RPE and RPE-Square in the URF addition. The training data are from the URF $4$-addition and the testing data are from the URF $5$-addition. (a) Both the models are trained for $20000$ steps. The comparison result shows that the RPE fails while the RPE-Square succeeds in achieving length generalization in the URF addition, (b) The RPE-Square model is trained for $200000$ steps, which shows the RPE-Square model converges to a solution achieving nearly perfect accuracy. The "$k$-th" line represents the digitwise accuracy to predict the $k$-th digit, i.e. $z_k$.

generalization, demonstrating the necessity of inductive bias for achieving length generalization. We further investigate the inductive biases of different model architectures trained with (S)GD in the context of Boolean functions. LDHD generalization can only be achieved if the inductive bias aligns with the structure of the target concept. As implications for length generalization in practical problems, our perspective on LDHD generalization elucidates the role of CoT in extending the latent space and leads to the principle that both inherent LDHD generalization and nuisances must be addressed to achieve length generalization.

For future work, while our theory is established with simplified models, it is crucial to further investigate inductive bias in more practical and complex models. Additionally, developing a paradigm for position embedding design based on the proposed principles is a valuable avenue for exploration.

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

## A    BACKGROUND ON BOOLEAN ANALYSIS

In this section, we include a brief background on Boolean analysis essential to this work. We refer to O'Donnell (2014) for further details and comprehensive coverage of Boolean analysis.

**Fourier Expansion.** A Boolean function $f : \{-1, 1\}^n \mapsto \mathbb{R}$ can always be represented by

$$f(x) = \sum_{T \subset [n]} \hat{f}(T)\chi_T(x), \tag{2}$$

where $\chi_T(x) = \pi_{i \in [T]} x_i$. The polynomial in (2) is called the Fourier expansion of the Boolean function $f$. The number $\hat{f}(T)$ is the Fourier coefficient of $f$ on $T$. The set $\{\chi_T(x)\}_{T \subset [n]}$ forms a basis, named Fourier basis, for the product space w.r.t the inner product defined by

$$\langle f, g \rangle = \mathbb{E}_{x \sim U(\{-1,1\}^n)} \left[ f(x)g(x) \right].$$

**Degree and Degree Profile.** The degree of a Boolean function $f$ is the degree of its Fourier expansion, which is a polynomial. The degree profile of $f$, denoted by $\mathrm{DegP}(f)$, is a $(n+1)$-tuple where $\mathrm{DegP}_i(f) = \sum_{T \subset [n], |T| = n+1-i} \hat{f}^2(T)$. The order between two degree profiles is lexicographic. The degree profile can be roughly seen as the distribution of the degrees of the monomials in the polynomial. Intuitively, the degree profile reflects the "complexity" of the Boolean function. A lower-degree-profile Boolean function uses fewer variables or combines them more simply than a higher-degree-profile one.

# B    INTUITIVE ILLUSTRATION OF LDHD GENERALIZATION

LDHD generalization is a special scenario of OOD generalization with structural distributional shifts. The unique structure captures the inherent scaling challenge in length generalization. Figure 2 is an intuitive comparison of in-distribution, (typical) OOD generalization, and LDHD generalization.

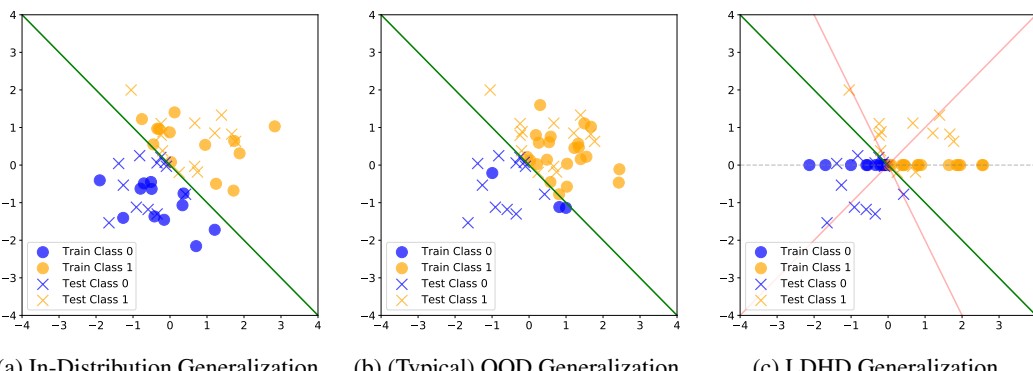

(a) In-Distribution Generalization.    (b) (Typical) OOD Generalization.    (c) LDHD Generalization.

Figure 2: Illustrative comparison between in-distribution generalization, typical out-of-distribution generalization (with distributional shift), and LDHD generalization. (a) For in-distribution generalization, the training distribution and the testing distribution are identical. (b) For typical OOD generalization, there is a shift between the training distribution and the testing distribution. The two distributions are somewhat "close" (e.g., sharing the same support set or having a small distributional distance). (c) For LDHD generalization, the training distribution and the testing distribution can be very different. While LDHD generalization can be seen as a type of OOD generalization, the shift has a special structure from a low-dimension subspace to a higher-dimension space. This makes LDHD generalization particularly challenging because the training data reveal no information about how the extra dimension contributes to the label. For example, suppose that we are learning a linear decision boundary from the low-dimension training data. We need to know the slope of the decision boundary as a prior; otherwise we will fail to achieve LDHD generalization. While all the three solid lines in (c) perfectly separate the training data, only the true decision boundary can achieve LDHD generalization.

# C    DETAILED EXPLANATION OF PLAA

In this section, we illustrate how PLAA and its variants abstract the effect of position embedding on the LDHD generalization of decoder-only Transformers.

## C.1    CONSTRUCTION OF PLAA

A typical linear attention at query $n$ can be expressed as

$$f_{\text{LA}}\left(x; W_Q, W_K, W_V, B\right) = \sum_{i \leq n} \left[\left(W_Q x_n\right)^\intercal \left(W_K x_i\right) + B_{i,n}\right] W_V x_i, \tag{3}$$

where $W_Q, W_K, W_V$ are the query, key, value matrices, respectively, and $B \in \mathbb{R}^{N \times N}$ is the position bias. To focus on the impact of the position embeddings, we fix $W_V = I$ and consider position-only attention score. Then we have

$$f_{\text{PLA}}\left(x; B\right) = \sum_{i \leq n} B_{i,n} x_i = \sum_{i \leq n} e_i^\intercal B e_n x_i.$$

Rewriting $B$ as $A$ and restricting $A$ to an upper triangular matrix, we have

$$f_{\text{PLA}}\left(x; A\right) = \sum_{i=1}^{n} e_i^\intercal A e_n x_i = x^\intercal A e_n, \quad \text{where } A \in \mathcal{U}_N.$$

In the computation of attention, the query position is asymmetric to other positions and can reflect the scale of the instance. To formalize the intuition, we introduce the notation $n(x) := \arg\max_{i \in [N]} \{x_i = -1\}$, which represents the last dimension where $x_i = -1$. In the LDHD generalization framework, $n(x)$ can be interpreted as the lowest dimension of the subspaces containing $x$, reflecting the dimension of $x$. We take $n(x)$ as the query position of $x$. Specifically, when $x$ is all-ones, it is treated as an "empty" sequence. In this case, we fix its output to align with the definition of PLAA. By the above derivation, we obtain Definition 5.

## C.2 Construction of PLAA with APE

The expression of the linear attention with APE is slightly different from (3). In practice, APE is commonly added to the token embeddings. Therefore, we omit the position bias and add APE to each $x_i$ when computing the attention score (we slightly abuse the notation of $x_i$ to denote the token embedding), i.e.,

$$f_{\text{LA}}^{\text{APE}}\left(x; W_Q, W_K, W_V, \{p_i\}_{i \in [N]}\right) = \sum_{i \leq n} [W_Q(x_n + p_n)]^\intercal [W_K(x_i + p_i)] W_V x_i,$$

where $p_i$ is the position embedding for the position $i$.

Similar to the derivation in Appendix C.1, we have

$$f_{\text{PLAA}}^{\text{APE}}(x; W_Q, W_K) = \sum_{i \leq n(x)} p_{n(x)}^\intercal W_Q^\intercal W_K p_i x_i = \sum_{i \leq n(x)} e_{n(x)}^\intercal P^\intercal W_Q^\intercal W_K P e_i x_i,$$

where $P$ is the learnable position embedding matrix such that $p_i = P e_i$ for all $i \in [N]$. Without loss of generality, we absorb the learnable parameters $W_Q, W_K$ into $P$, obtaining

$$f_{\text{PLAA}}^{\text{APE}}(x; P) = \sum_{i \leq n} x_i^\intercal e_i^\intercal P^\intercal P e_{n(x)} = x^\intercal \left(M_N^u \circ P^\intercal P\right) e_{n(x)}.$$

## C.3 Construction of PLAA with GRPE

RPE can be seen as a reparameterization of the position bias matrix such that the entries on the same diagonals share learnable parameters, i.e.,

$$B = \sum_{i=1}^N D_i p_i,$$

where $D_k$ is the $k$-th upper diagonal matrix and $p_i$ is the learnable parameter for all $1 \leq k \leq N$.

We generalize RPE so that our model can cover more general position biases. A natural extension is to choose general upper triangular matrices, denoted by $\mathcal{U}_{\text{GRPE}} = \{U_1, \ldots, U_r\}$, instead of $\mathcal{U}_{\text{RPE}} = \{D_1, \ldots, D_N\}$. To normalize, we suppose $\langle U_i, U_i \rangle = 1$ for all $i = 1, \ldots, r$. (This condition does not hold for $\mathcal{U}_{\text{RPE}}$. We can slightly modify the matrices by replacing $D_k$ with $\frac{1}{N+1-k} D_k$ to satisfy the condition.) We further require $(U_i)_{kl}(U_j)_{kl} = 0$ for all $i \neq j$ and $1 \leq k, l \leq N$, i.e., each $B_{i,j}$ is characterized by at most one learnable parameter. Using the reparameterization

$$B = \sum_{i=1}^r U_i p_i$$

in the derivation of PLAA, we obtain PLAA with GRPE in Definition 7.

# D PROOFS

## D.1 PROOF FOR THEOREM 1

Let $\mathcal{D}_1 := \mathcal{D}\left(\mathcal{X}_N \mid x \in \mathcal{X}_{N_0}\right)$ be the conditional distribution given that $x \in \mathcal{X}_{N_0}$ and $\mathcal{D}_2 := \mathcal{D}\left(\mathcal{X}_N \mid x \notin \mathcal{X}_{N_0}\right)$ be the conditional distribution given that $x \notin \mathcal{X}_{N_0}$

$$\sum_{f \in \mathcal{F}/\left(c_1|_{\mathcal{X}_{N_0}}\right)} \mathbb{E}_{x \sim \mathcal{D}(\mathcal{X}_N)}\left[\ell\left(c_1(x), f(x)\right)\right]$$

$$= P_{\mathcal{D}(\mathcal{X}_N)}(x \in \mathcal{X}_{N_0}) \sum_{f \in \mathcal{F}/\left(c_1|_{\mathcal{X}_{N_0}}\right)} \mathbb{E}_{x \sim \mathcal{D}_1}\left[\ell\left(c_1(x), f(x)\right)\right]$$

$$+ P_{\mathcal{D}(\mathcal{X}_N)}(x \notin \mathcal{X}_{N_0}) \sum_{f \in \mathcal{F}/\left(c_1|_{\mathcal{X}_{N_0}}\right)} \mathbb{E}_{x \sim \mathcal{D}_2}\left[\ell\left(c_1(x), f(x)\right)\right]$$

$$= P_{\mathcal{D}(\mathcal{X}_N)}(x \in \mathcal{X}_{N_0})V_1(c_1) + P_{\mathcal{D}(\mathcal{X}_N)}(x \notin \mathcal{X}_{N_0})V_2(c_1),$$

where

$$V_1(c) := \sum_{f \in \mathcal{F}/\left(c|_{\mathcal{X}_{N_0}}\right)} \mathbb{E}_{x \sim \mathcal{D}_1}\left[\ell\left(c(x), f(x)\right)\right],$$

$$V_2(c) := \sum_{f \in \mathcal{F}/\left(c|_{\mathcal{X}_{N_0}}\right)} \mathbb{E}_{x \sim \mathcal{D}_2}\left[\ell\left(c(x), f(x)\right)\right].$$

Similarly, we have

$$\sum_{f \in \mathcal{F}/\left(c_2|_{\mathcal{X}_{N_0}}\right)} \mathbb{E}_{x \sim \mathcal{D}(\mathcal{X}_N)}\left[\ell\left(c_2(x), f(x)\right)\right] = P_{\mathcal{D}(\mathcal{X}_N)}(x \in \mathcal{X}_{N_0})V_1(c_2) + P_{\mathcal{D}(\mathcal{X}_N)}(x \notin \mathcal{X}_{N_0})V_2(c_2).$$

To prove the theorem, it remains to show that $V_1(c_1) = V_1(c_2)$ and $V_2(c_1) = V_2(c_2)$.

By the definition of $\mathcal{F}/\left(c|_{\mathcal{X}_{N_0}}\right)$, we have

$$V_1(c_1) = \sum_{f \in \mathcal{F}/\left(c_1|_{\mathcal{X}_{N_0}}\right)} \mathbb{E}_{x \sim \mathcal{D}_1}\left[\ell\left(c_1(x), f(x)\right)\right]$$

$$= \sum_{f \in \mathcal{F}/\left(c_1|_{\mathcal{X}_{N_0}}\right)} \mathbb{E}_{x \sim \mathcal{D}_1}\left[\ell\left(c_1(x), c_1(x)\right)\right]$$

$$\overset{(a)}{=} \sum_{f \in \mathcal{F}/\left(c_2|_{\mathcal{X}_{N_0}}\right)} \mathbb{E}_{x \sim \mathcal{D}_1}\left[\ell\left(c_2(x), c_2(x)\right)\right]$$

$$= \sum_{f \in \mathcal{F}/\left(c_2|_{\mathcal{X}_{N_0}}\right)} \mathbb{E}_{x \sim \mathcal{D}_1}\left[\ell\left(c_2(x), f(x)\right)\right] = V_1(c_2),$$

where the equality (a) is due to that $c_1(x) = c_2(x)$ for all $x \in \mathcal{X}_{N_0}$.

By the no-free-lunch theorem (Wolpert & Macready, 1997; Wolpert, 2002) for $\mathcal{X}_N \setminus \mathcal{X}_{N_0}$, $\mathcal{Y}$, and $\mathcal{F}_{\mathcal{X}_N \setminus \mathcal{X}_{N_0}, \mathcal{Y}}$, we have

$$\sum_{f \in \mathcal{F}_{\mathcal{X}_N \setminus \mathcal{X}_{N_0}, \mathcal{Y}}} \mathbb{E}_{x \sim \mathcal{D}_2}\left[\ell\left(c_1(x), f(x)\right)\right] = \sum_{f \in \mathcal{F}_{\mathcal{X}_N \setminus \mathcal{X}_{N_0}, \mathcal{Y}}} \mathbb{E}_{x \sim \mathcal{D}_2}\left[\ell\left(c_2(x), f(x)\right)\right].$$

Then we have

$$
\begin{aligned}
V_2(c_1) &= \sum_{f \in \mathcal{F}/\left(c_1|_{\mathcal{X}_{N_0}}\right)} \mathbb{E}_{x \sim \mathcal{D}_2}\left[\ell\left(c_1(x), f(x)\right)\right] \\
&= \sum_{f \in \mathcal{F}_{\mathcal{X}_N \setminus \mathcal{X}_{N_0}, \mathcal{Y}}} \mathbb{E}_{x \sim \mathcal{D}_2}\left[\ell\left(c_1(x), f(x)\right)\right] \\
&= \sum_{f \in \mathcal{F}_{\mathcal{X}_N \setminus \mathcal{X}_{N_0}, \mathcal{Y}}} \mathbb{E}_{x \sim \mathcal{D}_2}\left[\ell\left(c_2(x), f(x)\right)\right] \\
&= \sum_{f \in \mathcal{F}/\left(c_2|_{\mathcal{X}_{N_0}}\right)} \mathbb{E}_{x \sim \mathcal{D}_2}\left[\ell\left(c_2(x), f(x)\right)\right] = V_2(c_2).
\end{aligned}
$$

### D.2 PROOF FOR THEOREM 2

Let $z = z(x) = \mathrm{Proj}(x, V)$ and $\mathcal{Z}_{N_0} = \{\mathrm{Proj}(x, V) \mid x \in \mathcal{X}_{N_0}\}$. Let $\mathcal{G}'_{N_0, c}$ be the set of all RFM interpolators on $\mathcal{Z}_{N_0}$ for the concept $c(z)$, i.e.,

$$
\mathcal{G}'_{N_0, c} = \{f_{\mathrm{RFM}}(z; a) \mid f_{\mathrm{RFM}}(z; a) = c(z) \text{ for all } z \in \mathcal{Z}_{N_0}\},
$$

where

$$
f_{\mathrm{RFM}}(z; a) = \frac{1}{\sqrt{K}} \sum_{k=1}^{K} a_k \sigma(\langle w_k, z \rangle + b_k).
$$

According to the proof of Theorem 3.8 in Abbe et al. (2023), the RFM model $f_{\mathrm{RFM}}(z; a)$ converges to the min-degree interpolator (w.r.t. the variable $z$) in $\mathcal{G}'_{N_0, c}$, i.e.,

$$
f_{\mathrm{RFM}}(z; a_t) \to \underset{g' \in \mathcal{G}'_{N_0, c}}{\arg \min} \, \mathrm{DegP}_{B'}(g'), \quad \text{as } K \to \infty, t \to \infty,
$$

where $B' = \{\chi_T(z)\}_{T \subseteq [r]}$.

By the definition of $z = z(x) = \mathrm{Proj}(x, V)$, we have

$$
f_{\mathrm{RFMP}}^{V, K}(x; a_t) = f_{\mathrm{RFM}}(z; a_t) \to \underset{g' \in \mathcal{G}'_{N_0, c}}{\arg \min} \, \mathrm{DegP}_{B'}(g') = \arg \underset{g \in \mathcal{G}_{N_0, c, V}}{\min} \, \mathrm{DegP}_{\mathcal{B}(V)}(g),
$$

as $K \to \infty, t \to \infty$.

### D.3 PROOF FOR COROLLARY 1

**Lemma 1.** *For any $f \in \mathcal{F}$, the min-degree interpolator $f^*$ on $\mathcal{X}_{N_0}$ satisfies $I(f^*) \subset [N_0]$.*

*Proof for Lemma 1.* Assume that there exist some $f \in \mathcal{F}$ such that the min-degreee interpolator $\hat{f}$ of $f$ on $\mathcal{X}_{N_0}$ does not satisfy $I(\hat{f}) \subset [N_0]$.

Let $\tilde{f}(x)$ be the function constructed by fixing all $x_i, i \notin [N_0]$ to 1 in $\hat{f}(x)$. By the construction, we have $\tilde{f}(x) = \hat{f}(x)$ for all $x \in \mathcal{X}_{N_0}$ and thus $\tilde{f}(x)$ is also an interpolator of $f$ on $\mathcal{X}_{N_0}$.

Since $I(\hat{f}) \not\subset [N_0]$, we have $\mathrm{DegP}(\tilde{f}) < \mathrm{DegP}(\hat{f})$. This contradicts the assumption that $\hat{f}$ is the min-degree interpolator. $\square$

By Lemma 1, for any $f \in \mathcal{F}$ such that $I(f) \not\subset [N_0]$, the min-degreee interpolator $\hat{f}$ is not identical to $f$ on $\mathcal{X}_N$ and thus does not achieves LDHD generalization from $\mathcal{X}_{N_0}$ to $\mathcal{X}_N$.

### D.4 PROOF FOR THEOREM 5

Since Theorem 3 is a special case of Theorem 5 for $\mathcal{U} = \{E_{ij}\}_{1 \leq i \leq j \leq N}$, we only need to prove Theorem 5.

Note that

$$f_{\text{PLAA}}^{\text{GRPE},\mathcal{U}}(x;p) = x^{\mathsf{T}} \left( \sum_{k=1}^{r} p_k U_k \right) e_{n(x)} = \left\langle xe_{n(x)}^{\mathsf{T}}, \sum_{k=1}^{r} p_k U_k \right\rangle.$$

Let $D_{N_0}$ be the size of the training set in $\mathcal{X}_{N_0}$. Then the loss can be represented as

$$L(p) = \frac{1}{2D_{N_0}} \sum_{x \in \mathcal{X}_{N_0}} \left( \left\langle xe_{n(x)}^{\mathsf{T}}, \sum_{k=1}^{r} p_k U_k \right\rangle - c^*(x) \right)^2.$$

Without loss of generality, we use the notation of gradient flow in this proof. Then we have

$$\dot{A} = -\frac{1}{D_{N_0}} \sum_{x \in \mathcal{X}_N} \left( \left\langle xe_{n(x)}^{\mathsf{T}}, A \right\rangle - c^*(x) \right) \sum_{k=1}^{r} \left\langle xe_{n(x)}^{\mathsf{T}}, U_k \right\rangle U_k.$$

Since $p(0) = 0$, we have $A(0) = 0$ and thus $A(t) \in \text{span} \left\{ \sum_{k=1}^{r} \left\langle xe_{n(x)}^{\mathsf{T}}, U_k \right\rangle U_k \right\}_{x \in \mathcal{X}_{N_0}}$. Then the convergence point $\hat{A}$ can be represented as $\hat{A} = \sum_{x \in \mathcal{X}_{N_0}} \hat{a}(x) \sum_{k=1}^{r} \left\langle xe_{n(x)}^{\mathsf{T}}, U_k \right\rangle U_k$. The convergence function $\hat{f}$ is

$$\hat{f}(x) = \sum_{x' \in \mathcal{X}_{N_0}} \hat{a}(x') \sum_{k=1}^{r} \left\langle x'e_{n(x')}^{\mathsf{T}}, U_k \right\rangle \left\langle xe_{n(x)}^{\mathsf{T}}, U_k \right\rangle$$

$$= \sum_{k=1}^{r} \sum_{x' \in \mathcal{X}_{N_0}} \hat{a}(x') \left\langle x'e_{n(x')}^{\mathsf{T}}, U_k \right\rangle \left\langle xe_{n(x)}^{\mathsf{T}}, U_k \right\rangle$$

**Lemma 2.** *For any upper triangle matrix $U \in \mathbb{R}^{N \times N}$, we have*

$$\langle xe_{n(x)}^{\mathsf{T}}, U \rangle = \sum_{1 \leq i \leq j \leq N} U_{ij} b_{ij}^{PLAA}(x).$$

*Proof for Lemma 2.* We first prove that $\langle xe_{n(x)}^{\mathsf{T}}, E_{mn} \rangle = b_{ij}^{\text{PLAA}}(x)$ for any $1 \leq i \leq j \leq N$. Notice that

$$\langle xe_{n(x)}^{\mathsf{T}}, E_{ij} \rangle = \begin{cases} x_i & x_j = -1 \wedge x_{j+1} = 1 \wedge \cdots \wedge x_N = 1, \\ 0 & \text{otherwise.} \end{cases}$$

Therefore, we have

$$\langle xe_{n(x)}^{\mathsf{T}}, E_{ij} \rangle = x_i \cdot \frac{1 - x_j}{2} \cdot \frac{1 + x_{j+1}}{2} \cdot \cdots \cdot \frac{1 + x_N}{2}$$

$$= \begin{cases} -\frac{1-x_j}{2} \prod_{k=j+1}^{N} \frac{1+x_k}{2}, & i = j, \\ x_i \frac{1-x_j}{2} \prod_{k=j+1}^{N} \frac{1+x_k}{2}, & i < j. \end{cases}$$

$$= b_{ij}^{\text{PLAA}}(x).$$

For any upper triangle matrix $U = \sum_{1 \leq i \leq j \leq N} U_{ij} E_{ij}$, we have

$$\langle xe_{n(x)}^{\mathsf{T}}, U \rangle = \langle xe_{n(x)}^{\mathsf{T}}, \sum_{1 \leq i \leq j \leq N} U_{ij} E_{ij} \rangle = \sum_{1 \leq i \leq j \leq N} U_{ij} \langle xe_{n(x)}^{\mathsf{T}}, E_{ij} \rangle = \sum_{1 \leq i \leq j \leq N} U_{ij} b_{ij}^{\text{PLAA}}(x).$$

$\square$

Define $b_k(x) := \langle xe_{n(x)}^{\mathsf{T}}, U_k \rangle$. By Lemma 2, we have $b_k(x) = \sum_{1 \le i \le j \le N} (U_k)_{ij} b_{ij}^{\text{PLAA}}(x)$. Then the convergence function $\hat{f}$ can be represented as

$$\hat{f}(x) = \sum_{k=1}^{r} \sum_{x' \in \mathcal{X}_{N_0}} \hat{a}(x') b_k(x') b_k(x) = \sum_{k=1}^{r} \hat{p}_k b_k(x),$$

where $\hat{p}_k = \sum_{x' \in \mathcal{X}_{N_0}} \hat{a}(x') b_k(x')$.

For any interpolator $g(x) = \sum_{k=1}^{r} p_k b_k(x)$ on $\mathcal{X}_{N_0}$:

- If $b_k(x) = 0$ for all $x \in \mathcal{X}_0$, we have $\hat{p}_k = 0$ and $p_k^2 \ge 0 = \hat{p}_k^2$;
- If $b_k(x) \ne 0$ for some $x \in \mathcal{X}_0$, we have $\hat{p}_k = p_k$ and thus $\hat{p}_k^2 = p_k^2$. To show this, without loss of generality, we suppose that for $1 \le k \le m$, there exists some $x \in \mathcal{X}_{N_0}$ such that $b_k(x) \ne 0$, and for $m+1 \le k \le r$, $b_k(x) = 0$ for all $x \in \mathcal{X}_{N_0}$. It suffices to show that the following equation has a unique solution:
$$\begin{bmatrix} b_1(x_1) & \dots & b_m(x_1) \\ \vdots & \vdots & \vdots \\ b_1(x_M) & \dots & b_m(x_M) \end{bmatrix} \begin{bmatrix} p_1 \\ \vdots \\ p_m \end{bmatrix} = \begin{bmatrix} c^*(x_1) \\ \vdots \\ c^*(x_M) \end{bmatrix},$$
where $M = |\mathcal{X}_{N_0}|$ and $\mathcal{X}_{N_0} = \{x_1, \dots, x_M\}$. Since $p_1 = \hat{p}_1, \dots, p_m = \hat{p}_m$ is a solution, it remains to prove the uniqueness. Let
$$B_m = \begin{bmatrix} b_1(x_1) & \dots & b_m(x_1) \\ \vdots & \vdots & \vdots \\ b_1(x_M) & \dots & b_m(x_M) \end{bmatrix}.$$

It suffices to show $\text{rank}(B_m) = m$. Note that $(B_m^{\mathsf{T}} B_m)_{ii} = \sum_{x \in \mathcal{X}_{N_0}} b_i(x)^2 > 0$ and $(B_m^{\mathsf{T}} B_m)_{ij} = \sum_{x \in \mathcal{X}_{N_0}} b_i(x) b_j(x) = 0$. We have $\text{rank}(B_m^{\mathsf{T}} B_m) = m$ and thus $\text{rank}(B_m) = m$.

Hence, for any interpolator $g$ on $\mathcal{X}_{N_0}$, we have

$$\text{DegP}_{\mathcal{B}_{\text{PLAA}}^{\text{GRPE},\mathcal{U}}}(\hat{f}) \le \text{DegP}_{\mathcal{B}_{\text{PLAA}}^{\text{GRPE},\mathcal{U}}}(g),$$

or equivalently,

$$f_{\text{PLAA}}^{\text{GRPE},\mathcal{U}}(x; p_t) \to \arg\min_{g \in \mathcal{G}_{N_0, p^*}^{\text{GRPE},\mathcal{U}}} \text{DegP}_{\mathcal{B}_{\text{PLAA}}^{\text{GRPE},\mathcal{U}}}(g), \quad \text{as } t \to \infty.$$

### D.5 PROOF FOR COROLLARY 3

By Theorem 5, the PLAA with GRPE achieves LHDH generalization from $\mathcal{X}_{N_0}$ to $\mathcal{X}_N$ if and only if the target concept $c(x)$ is the min-degree interpolator w.r.t. the linearly independent set $\mathcal{B}_{\text{PLAA}}^{\text{GRPE},\mathcal{U}}$. Therefore, it is equivalent to prove the target concept $c(x)$ is the min-degree interpolator w.r.t. the linearly independent set $\mathcal{B}_{\text{PLAA}}^{\text{GRPE},\mathcal{U}}$ if and only if

$$\left\{ k \mid (U_k)_{[N_0],[N_0]} = 0 \right\} \subseteq \left\{ k \mid c_k = 0 \right\}.$$

We denote $\left\{ k \mid (U_k)_{[N_0],[N_0]} = 0 \right\}$ by $\mathcal{K}_1$ and $\left\{ k \mid c_k = 0 \right\}$ by $\mathcal{K}_2$ in this proof.

We first show the sufficiency. Assume that $c(x)$ is not the min-degree interpolator w.r.t. the linearly independent set $\mathcal{B}_{\text{PLAA}}^{\text{GRPE},\mathcal{U}}$ on $\mathcal{X}_{N_0}$. In other words, there exists an interpolator $\tilde{c}(x) = \sum_{k=1}^{r} \tilde{c}_k \sum_{1 \le i \le j \le N} (U_k)_{ij} b_{ij}^{\text{PLAA}}(x)$ on $\mathcal{X}_{N_0}$ such that

$$\text{DegP}_{\mathcal{B}_{\text{PLAA}}^{\text{GRPE},\mathcal{U}}}(\tilde{c}) < \text{DegP}_{\mathcal{B}_{\text{PLAA}}^{\text{GRPE},\mathcal{U}}}(c).$$

Since both $c(x)$ and $\tilde{c}(x)$ are interpolators on $\mathcal{X}_{N_0}$, we have

$$\sum_{k=1}^{r} c_k (U_k)_{[N_0],[N_0]} = \sum_{k=1}^{r} \tilde{c}_k (U_k)_{[N_0],[N_0]},$$

or equivalently,

$$\sum_{k \notin \mathcal{K}_1} c_k (U_k)_{[N_0],[N_0]} = \sum_{k \notin \mathcal{K}_1} \tilde{c}_k (U_k)_{[N_0],[N_0]}.$$

Since $(U_i)_{kl}(U_j)_{kl} = 0$ for all $i \neq j$ and $1 \leq k, l \leq N$, we have $\left\{ (U_k)_{[N_0],[N_0]} \right\}_{k \notin \mathcal{K}}$ are linearly independent. This implies $c_k = \tilde{c}_k$ for all $k \notin \mathcal{K}_1$.

Since $\mathcal{K}_1 \subseteq \mathcal{K}_2$, we have $c_k = 0$ and thus $c_k^2 \leq \tilde{c}_k^2$ for all $k \in \mathcal{K}_1$. Hence, we have

$$\mathrm{DegP}_{\mathcal{B}_{\mathrm{PLAA}}^{\mathrm{GRPE},\mathcal{U}}}(c) \leq \mathrm{DegP}_{\mathcal{B}_{\mathrm{PLAA}}^{\mathrm{GRPE},\mathcal{U}}}(\tilde{c}),$$

which contradicts the assumption.

We then prove the necessity. Assume that $\mathcal{K}_1 \nsubseteq \mathcal{K}_2$. Then there exists some $k_0 \in \mathcal{K}_1$ but $k_0 \notin \mathcal{K}_2$. Define $\tilde{\tilde{c}}(x) = \sum_{k=1}^r \tilde{\tilde{c}}_k \sum_{1 \leq i \leq j \leq N} (U_k)_{ij} b_{ij}^{\mathrm{PLAA}}(x)$ where

$$\tilde{\tilde{c}}_k = \begin{cases} 0, & k = k_0, \\ c_k, & k \neq k_0. \end{cases}$$

By the definition of $\mathcal{K}_1$, $\tilde{\tilde{c}}(x) = c(x)$ for all $x \in \mathcal{X}_{N_0}$ and thus $\tilde{\tilde{c}}(x)$ is also an interpolator on $\mathcal{X}_0$. Note that $c_{k_0} \neq 0$. By the definition of $\tilde{\tilde{c}}(x)$, we have

$$\mathrm{DegP}_{\mathcal{B}_{\mathrm{PLAA}}^{\mathrm{GRPE},\mathcal{U}}}(\tilde{\tilde{c}}) \leq \mathrm{DegP}_{\mathcal{B}_{\mathrm{PLAA}}^{\mathrm{GRPE},\mathcal{U}}}(c),$$

which contradicts that the target concept $c(x)$ is the min-degree interpolator w.r.t. the linearly independent set $\mathcal{B}_{\mathrm{PLAA}}^{\mathrm{GRPE},\mathcal{U}}$ on $\mathcal{X}_{N_0}$.

# E  EXPERIMENTS

## E.1  DETAILED EXPLANATION OF RPE-SQUARE

The design of RPE-Square follows the principle: when devising position embeddings for length generalization, one needs to consider both the LDHD generalization in the latent space and the nuisance of the data format in the input sequence space. This principle is derived from the LDHD generalization perspective for length generalization, illustrating the insights of LDHD generalization for practical models.

To further elaborate, we show how RPE-Square is constructed in the guidance of the proposed principle to achieve length generalization of the URF addition (with CoT).

- **LDHD generalization in the latent space**. For the addition, the LDHD generalization in the latent space can be handled by RPE.

- **The data format nuisance of URF**. To handle the data format, we need to consider the mapping between a latent variable and its URF string, i.e., how the latent variable $h_n = [(x_0, y_0, z_0), \ldots, (x_{t-1}, y_{t-1}, z_{t-1}), (x_t, y_t, *), \ldots, (x_{n-1}, y_{n-1}, *)]$ can be recovered from the corresponding URF string "$\texttt{[BOS]}\texttt{x}_0 \ldots \texttt{x}_{\texttt{n}_x-1} + \texttt{y}_0 \ldots \texttt{y}_{\texttt{n}_y-1} = \texttt{z}_0 \ldots \texttt{z}_{t-1}$" (to predict $\texttt{z}_\texttt{t}$). The URF mapping can be described as follows: concatenate "$\texttt{[BOS]}$", the first elements of the dimensions (up to the "highest" nonzero element), a "$+$", the second elements of the dimensions (up to the "highest" nonzero element), a "$=$", and the third elements of the dimensions (up to the "highest" non-"$*$" element).

  According to the URF mapping, we notice that the "position" of an element in the latent variable can be identified by its relative distances to "$\texttt{[BOS]}$", "$+$", and "$=$". (Here, the relative distance from the position $i$ to the position $j$ is $i - j$.). Denote the tuple of the relative distances from some token $s$ to "$\texttt{[BOS]}$", "$+$", "$=$" by $(n_1(s), n_2(s), n_3(s))$. Then we have

  $$s = \begin{cases} x_{n_1(s)-1}, & n_1(s) > 0, n_2(s) < 0, n_3(s) < 0, \\ y_{n_2(s)-1}, & n_1(s) > 0, n_2(s) > 0, n_3(s) < 0, \\ z_{n_3(s)-1}, & n_1(s) > 0, n_2(s) > 0, n_3(s) > 0. \end{cases}$$

Note that $z_k$ can be determined by $x_{k-1}, x_k, y_{k-1}, y_k$, and $z_k$ (we ignore the boundary cases in this discussion for simplicity). To predict the next token of some $s$ where $n_1(s) > 0, n_2(s) > 0, n_3(s) > 0$, we need to identify the elements $s_1, s_2, s_3, s_4, s_5$ such that

$$\begin{cases} n_1(s_1) = n_3(s) - 1, & n_2(s_1) < 0, & n_3(s_1) < 0, \\ n_1(s_2) = n_3(s), & n_2(s_2) < 0, & n_3(s_2) < 0, \\ n_2(s_3) = n_3(s) - 1, & n_1(s_3) > 0, & n_3(s_3) < 0, \\ n_2(s_4) = n_3(s), & n_1(s_4) > 0, & n_3(s_4) < 0, \\ n_3(s_5) = n_3(s) - 1, & n_1(s_5) > 0, & n_2(s_5) > 0. \end{cases}$$

Therefore, to address the URF data format nuisance, the position embedding can consider the relative distances to some tokens.

RPE-Square is designed to encode the prior knowledge of both the LDHD generalization and the data format, combining RPE and the approach to deal with the URF data format. The position embedding for the query $j$ and the key $i$ is determined by *the relative distance* between *the relative distance of $j$ to the position of some token $s_l$*, and *the relative distance of $i$ to the position of some token $x_k$*, parameterized by $R_{(j-l)-(i-k)}$. The "relative distance of relative distances" design is where the name RPE-Square comes from. Furthermore, we want to learn the special tokens automatically rather than specify them by hand-craft. To achieve this, we adopt a "soft" design. taking a weighted average over all $1 \leq l \leq j, 1 \leq k \leq i$. The weight of $R_{(j-l)-(i-k)}$ depends on the tokens, which is the multiplication between the attention score of the query $x_j$ and the key $x_l$, and the attention score of the query $x_i$ and the key $x_k$. This design makes RPE-Square more flexible and potentially applicable to other problems of structures similar to the URF addition. We conduct an addition experiment in Appendix E.3, showing that RPE-Square can achieve length generalization beyond the URF addition.

## E.2 EXPERIMENT DETAILS

For the URF $n$-addition training data, we first sample the lengths of two addends uniformly from $\{1, \ldots, n\} \times \{1, \ldots, n\}$. For two addends of lengths $(n_1, n_2)$, we then samples from $[10^{n_1-1}, 10^{n_1} - 1] \times [10^{n_2-1}, 10^{n_2} - 1]$ (If $n_i = 0$, then the corresponding sample interval is $[0, 9]$). This is to guarantee the addends are length-uniform. For the addition $x_{n_1-1} \ldots x_0 + y_{n_2-1} \ldots y_0 = z_{n_3} z_{n_3-1} \ldots z_0$, the training instance is

$$\mathtt{b}\, \mathtt{x_0} \, \ldots \, \mathtt{x_{n_1-1}} \, + \, \mathtt{y_0} \, \ldots \, \mathtt{y_{n_2-1}} \, = \, \mathtt{z_0} \, \ldots \, \mathtt{z_{n_3-1}} \, \mathtt{z_{n_3}} \, \mathtt{e}.$$

Here, we add spaces between the characters to ensure each is tokenized separately. We use "b" and "e" instead of "[BOS]" and "[EOS]" for a simpler implementation with the GPT-2 tokenizer. In our experiments, we train with $10000$ URF 4-addition samples.

We choose GPT-2 with key-only position embeddings as our model. For the RPE and RPE-Square settings, we augment the GPT-2 model with RPE and RPE-Square, respectively. The implementation is adapted from HuggingFace (Wolf et al., 2020).

We train the models by AdamW, with the initial learning rate $0.0005$, the weight decay $1.0$, and the cosine scheduler. The warmup ratio is $0.05$. The gradient accumulation steps is $2$. The per-device training batch size is $128$. We train the models for $20000$ steps and $200000$ steps. The experiments are run on a server with Ubuntu. The models are trained on two NVIDIA GeForce RTX 3090 GPUs.

## E.3 ADDITIONAL EXPERIMENTS

To further validate the LDHD generalization perspective, we additionally consider the (unaligned) copy. We show that RPE-Square, the position embedding derived according to the LDHD generalization perspective, can achieve length generalization in this task, while RPE fails.

Concretely, an instance of the copy task is like

$$\mathtt{b}\, \mathtt{x_0} \, \ldots \, \mathtt{x_{n-1}} \, = \, \mathtt{x_0} \, \ldots \, \mathtt{x_{n-1}} \, \mathtt{e}.$$

The model is given the input $\mathtt{bx_0 \ldots x_{n-1}} =$ and expected to output the copy of $\mathtt{x_0 \ldots x_{n-1}}$.

We sample $2000$ $n$-length instances for each $n = 1, \ldots, 5$ as the training data. In the evaluation, we examine the learned models on instances of length $1 - 10$. We train GPT-2 with key-only RPE

and RPE-Square, respectively. The model is trained by AdamW with the cosine scheduler, where the initial learning rate is $0.0005$, the weight decay is $1.0$, the warmup ratio is $0.05$, the gradient accumulation step is $2$, and the per-device training batch size is $256$. We set the training steps to $10000$ but early stop at step $1000$ as the model with RPE-Square has achieved nearly perfect length generalization while the model with RPE shows almost no length generalization then. The result is presented in Figure 3.

From the perspective of LDHD generalization, the latent variable corresponding to the input $\mathtt{bx_0 \ldots x_{n-1} = x_0 \ldots x_k}$ is

$$[(x_0, x_0), \ldots, (x_k, x_k), (x_{k+1}, *), \ldots, (x_N, x_N)].$$

The LDHD generalization in the latent space can be effectively addressed by RPE. The data format mapping can be handled by considering the relative distance to the tokens "b" and "=". Therefore, RPE-Square is expected to work for the length generalization of the unaligned copy. RPE does not properly deal with the unaligned data format and thus could fail to achieve length generalization in this scenario.

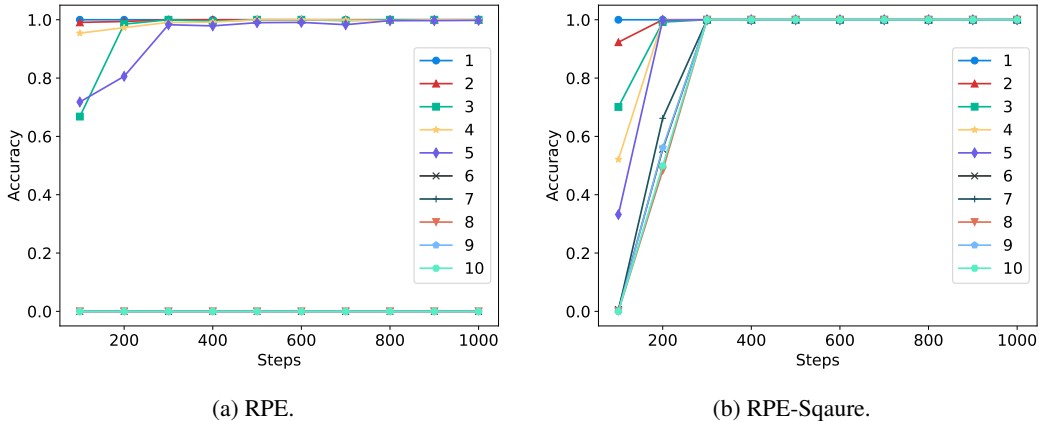

(a) RPE.  (b) RPE-Sqaure.

Figure 3: Length generalization of Transformer with RPE (a) and RPE-Square (b) in the unaligned copy. The training data are of length ranging from 1 to 5. Both the models are trained for 1000 steps. We evaluate the models on data of length ranging from 1 to 5. While both the models have good in-distribution generalization performance, only the model with RPE-Square generalizes to the instances of length 6-10.

