# OpenReview forum: "Low-Dimension-to-High-Dimension Generalization and Its Implications for Length Generalization"
_ICLR.cc/2025/Conference — Submitted to ICLR 2025_

### Official Review · Reviewer_eFtA · 2024-11-02

**Soundness:** 4
**Presentation:** 1
**Contribution:** 2
**Rating:** 3
**Confidence:** 3

**Summary:**

This paper is about a notion similar to domain adaption, but tailored to scenarios found in large language models, that the authors named low-dimension-to-high-dimension (LDHD) generalization. The authors give generic impossibility results in the style of the No-Free-Lunch theorem, and also specific impossibility results for several models, drawing attention to the issue that LDHD cannot happen without inductive bias.

**Strengths:**

This is an important problem. The authors made commendable efforts in understanding LDHD for Boolean functions and connections to the implicit regularization line of research to decipher what kind of inductive may or may not be helpful for LDHD.

**Weaknesses:**

The main issue for me is with the presentation and the writing. There is a big conceptual gap between the introduction, especially the two examples given involving addition, with the rest of the paper. After reading examples 1 and 2, I still don't have sense of the relevance toward LDHD that they are intending to convey. Due to this, I find the introduction to be difficult to understand.

It is not clear how the notion of LDHD relates to previous works on domain adaptation e.g., [1]. The only novelty in Definition 1 seems to be that the two domains are nested and both are included inside a larger ambient space. This scenario can be captured by previous notions of OOD generalization.

Given that Boolean functions play an important role, there should be more discussions about what is degree, what is the intuition behind degree profiles.

Assumption 2 has been refuted in [2]. While this doesn't invalidate results from Sec 4.2, I feel that this should be discussed.

[1] David, Shai Ben, et al. "Impossibility theorems for domain adaptation." Proceedings of the Thirteenth International Conference on Artificial Intelligence and Statistics. JMLR Workshop and Conference Proceedings, 2010.

[2] Li, Zhiyuan, Yuping Luo, and Kaifeng Lyu. "Towards Resolving the Implicit Bias of Gradient Descent for Matrix Factorization: Greedy Low-Rank Learning." International Conference on Learning Representations.

**Questions:**

More specifically, I think the authors should address these following points to make the introduction more understandable:

L052: I don't understand Example 1. How does this example illustrate LDHD?
L062: What does relative position embedding have to do with the main thesis of this paper? I find this remark to be distracting.

L076: What is a "concept"? How does this show up in Example 1 and 2? Neither example 1 nor 2 have a label. Also, the notation phi is completely new. If the examples are meant to motivate Assumption 1, it should be made clear.

L088: Why is Sigma a field now? In example 1 and 2, Sigma is the set of tuple of digits.

L096: How is the No-Free Lunch theorem relevant here? The No-Free Lunch Theorem says that there's no learning algorithm is can be optimal for all problem instances. I don't see the relevance to LDHD.

L102: What is "minimal degree-profile bias"?
L107: What are "independent sets"? Do you mean "linearly independent"?

---

> ### Author Response · Authors · 2024-11-20
>
> Thank you for your valuable comments. Here are our responses.
>
> **Response to Weaknesses:**
>
> 1. *There is a big conceptual gap between the introduction, especially the two examples given involving addition, with the rest of the paper. After reading examples 1 and 2, I still don't have sense of the relevance toward LDHD that they are intending to convey.*
>
>
> Example 1 demonstrates the motivation to consider LDHD generalization.
> Example 2 demonstrates the intuition behind LDHD generalization.
>
> Example 1 illustrates why string length does not faithfully reflect the scale of the problem in addition.
> More specifically, in ARF and URF, a smaller-scale addition instance can have the same length or even larger length than a larger-scale one.
> This mismatch is because the string length is affected by both the scale and the data format.
>
> Example 2 illustrates the typical length generalization in addition and motivates the LDHD perspective for length generalization. More concretely, in addition, an $n$-addition can be represented by an $n$-dimension state vector, where each dimension indicates one digit of the two addenda. The scale of an addition instance is reflected by the dimension of the state vector.
>
> The line of the introduction goes as follows:
> 1) String length does not precisely capture the scaling challenge in length generalization.
> 2) Illustrate 1) by Example 1.
> 3) Due to 1), a new formulation of length generalization is required. But how?
> 4) Illustrate the intuition of length generalization in practice by Example 2.
> 5) According to the intuition, we then propose our LDHD framework to characterize the problem scaling.
>
>
> 2. *It is not clear how the notion of LDHD relates to previous works on domain adaptation e.g., [1]. The only novelty in Definition 1 seems to be that the two domains are nested and both are included inside a larger ambient space. This scenario can be captured by previous notions of OOD generalization.*
>
> The novelty of LDHD generalization cannot be captured by domain adaptation or OOD generalization.
> - LDHD generalization is a special case of unseen generalization, which is focused on the generalization to the instances that are not in the support set of the training distribution. While unseen generalization can be seen as a type of OOD generalization, the general notion of OOD generalization is insufficient to capture the challenge of unseen generalization. Particularly, while previous works on domain adaptation typically assume the training distribution and the testing distribution are somewhat "close" (e.g., small distance), unseen generalization makes no such assumptions.
> - LDHD generalization cannot be fully captured by the notation unseen generalization, either. The main point of LDHD generalization is to characterize a family of unseen generalization with a special structure, i.e., the testing instances can be of higher dimension than the training ones. This structure captures the inherent scaling challenge in length generalization.
>
> 3. *Given that Boolean functions play an important role, there should be more discussions about what is degree, what is the intuition behind degree profiles.*
>
> The degree of a Boolean function is the degree of the Fourier expansion of the Boolean function, a polynomial.
> The degree profile can be roughly seen as the distribution of the degrees of the monomials in the polynomial.
> Intuitively, the degree profile reflects the "complexity" of the Boolean function. A lower-degree-profile Boolean function uses fewer variables or combines them more simply than a higher-degree-profile one.
>
> In our work, we generalize the concept of the degree profile by defining it with respect to a set of linearly independent polynomials that are not necessarily the Fourier basis. Using this generalized degree profile, we demonstrate how different models converge to different Boolean functions, thereby determining whether they achieve LDHD generalization.
>
> We appreciate your advice and will provide more background on Boolean analysis in our updated manuscript.

---

> > ### Author Response · Authors · 2024-11-20
> >
> > 4. *Assumption 2 has been refuted in [2]. While this doesn't invalidate results from Sec 4.2, I feel that this should be discussed.*
> >
> > Some works propose that rank minimization provides a more accurate description than nuclear norm minimization under certain technical assumptions deemed more reasonable by the respective authors.
> > While the nuclear norm minimization and the rank minimization are not always equivalent, both can explain the failure of APE in length generalization in certain scenarios.
> > Furthermore, in the literature on low-rank optimization, the nuclear norm has been shown to serve as an effective surrogate for rank minimization under particular assumptions [1] [2] [3], making it a widely used regularization technique in practical applications.
> >
> > In this work, we focus on the length generalization problem and assume Conjecture 1 in this work for simplicity. A technical discussion on rank minimization and nuclear norm minimization is an important problem of independent interest.
> > We are grateful for your suggestion and would like to include the discussion to enhance the rigor of the manuscript.
> >
> > [1] Recht, Benjamin, Weiyu Xu, and Babak Hassibi. "Necessary and sufficient conditions for success of the nuclear norm heuristic for rank minimization." In 2008 47th IEEE Conference on Decision and Control, pp. 3065-3070. IEEE, 2008.
> >
> > [2] Candès, Emmanuel J., and Terence Tao. "The power of convex relaxation: Near-optimal matrix completion." IEEE transactions on information theory 56, no. 5 (2010): 2053-2080.
> >
> > [3] Candes, Emmanuel, and Benjamin Recht. "Exact matrix completion via convex optimization." Communications of the ACM 55, no. 6 (2012): 111-119.

---

> > > ### Author Response · Authors · 2024-11-20
> > >
> > > **Response to Questions:**
> > >
> > > **L052:** *I don't understand Example 1. How does this example illustrate LDHD?*
> > >
> > > Example 1 is the motivation for why we consider LDHD generalization rather than direct string length generalization.
> > > It illustrates that string length can fail to reflect the scale of the instances. More concretely, in ARF and URF, a smaller-scale addition instance can have the same length or even larger length than a larger-scale one.
> > > Furthermore, string length is affected by the data format and methods considering string length generalization can be sensitive to the data format. This means string length is not a good choice on which we define length generalization.
> > >
> > > **L076:** *What is a "concept"? How does this show up in Example 1 and 2? Neither example 1 nor 2 have a label. Also, the notation phi is completely new. If the examples are meant to motivate Assumption 1, it should be made clear.*
> > >
> > >
> > > A concept is a target mapping. The term "concept" is used in the description of the PAC model [1] and we follow the notation. In Assumption 1, the concept is the target function that determines the label from the latent variable. We define the concept on the latent space rather than the input space (e.g., the token sequences in language models) because we want it to be independent of the data format and represent the "core" mapping (e.g., the addition operator).
> > > In Examples 1-2, the concept is the mapping from the state vector to the sum of the corresponding two addenda or the mapping from the state vector to one next digit of the sum if the prediction is digit-by-digit as in autoregressive language models.
> > >
> > > The notation $\phi$ is the data format mapping. In the addition, for example, it can be seen the mapping from the latent state vector to the ARF or URF representations.
> > >
> > > [1] Mohri, Mehryar. "Foundations of machine learning." (2018), Chapter 2, The PAC Learning Framework, Section 2.1, The PAC learning model.
> > >
> > >
> > > **L088:** *Why is Sigma a field now? In example 1 and 2, Sigma is the set of tuple of digits.*
> > >
> > > Thank you for your observation regarding the term "field". In the manuscript, we used the term "field" informally to refer to the domain of the state vector. Since this terminology is improper in the strict mathematical sense and our results do not depend on the properties of field, we would like to revise it, using "domain" instead of "field", to avoid misinterpretation.
> > >
> > >
> > > **L096:** *How is the No-Free Lunch theorem relevant here? The No-Free Lunch Theorem says that there's no learning algorithm is can be optimal for all problem instances. I don't see the relevance to LDHD.*
> > >
> > >
> > > The No-Free-Lunch Theorem of LDHD generalization indicates that there is no learning algorithm that can achieve LDHD generalization for all problems. This result is due to LDHD. Intuitively, even if we have perfect knowledge of the training space, i.e., all instances in the training space, we have no information about some instances in the testing space because we do not know how to handle the extra dimensions in LDHD. In Theorem 1 (No-Free-Lunch Theorem of LDHD Generalization), $c_1$ and $c_2$ can be seen as the output of two algorithms given $\mathcal{X}_{N_0}$. The average losses of $c_1$ and $c_2$ over all potential targets $f$ (consistent with all the training data, i.e., $c_1(x)=c_2(x)=f(x)$ on $\mathcal{X}_0$) are always equal. In other words, no algorithm can be universally optimal for the LDHD generalization of all problems.
> > >
> > > The No-Free-Lunch Theorem of LDHD generalization describes an inherent difficulty of LDHD. As a result, we need prior knowledge to achieve length generalization for a given problem. This motivates our analysis of how different models converge to different Boolean functions with gradient descent, which illustrates how we can encode prior knowledge for length generalization via models and training algorithms jointly.
> > >
> > >
> > > **L102:** *What is "minimal degree-profile bias"?*
> > >
> > > Minimal degree-profile bias means that a model converges to the solution of the minimal degree profile, among all the interpolators of the training data. It is an inductive bias of the model design and the training algorithm (gradient descent). Intuitively, minimal degree-profile (w.r.t. a linearly independent set) reflects the complexity of a Boolean function (w.r.t. a model).
> > > Minimal degree-profile bias (w.r.t. a linearly independent set) means that the training algorithm tends to choose the "simplest" solution (w.r.t. the model).
> > >
> > > **L107:** *What are "independent sets"? Do you mean "linearly independent"?*
> > >
> > > Yes, "independent set" means "linearly independent set". We are sorry for your confusion and will use the term "linearly independent set" instead of "independent set".

---

> > > > ### Author Response · Authors · 2024-12-01
> > > > **End of Discussion Period Approaching**
> > > >
> > > > We sincerely appreciate your constructive feedback on our manuscript. As the end of the discussion period is approaching, we would be grateful if we could hear your feedback regarding our responses to your comments. Please let us know if you have further questions or suggestions—we would be happy to address them.

---

### Official Review · Reviewer_w7k4 · 2024-11-04

**Soundness:** 3
**Presentation:** 2
**Contribution:** 2
**Rating:** 6
**Confidence:** 2

**Summary:**

This paper frames the length generalization problem in tasks like addition as a low-dimensional to high-dimensional (LDHD) generalization problem. With this motivation, the paper studies LDHD generalization in two theoretical setups on Boolean functions, with Boolean input sequences of length $N$. Here, the data dimensionality is controlled by the number of final positions fixed to 1—that is, sequences coming from a low-dimensional space of size  $N_0 < N$ have the last $N - N_0$ positions set to 1.

In the first setup, the authors consider a random feature model *with projection* (RFMP), where the inputs are first projected into a subspace defined by matrix $V$ before being passed through an RFM. Extending the result in Abbe et al. (2023), they show gradient descent (GD) converges to a minimum degree interpolator, now with respect to a basis that depends on the choice of $V$. They use this result to argue that if the ground truth Boolean function depends on the higher dimensions of the input, the min-norm interpolator of the low-dimensional data will fail to capture it.

The second setup considers variations of linear attention models, simplified to focus on the role of positional embedding. Here, they show that different positional embedding changes the inductive bias of GD by changing the bases with respect to which the min degree interpolator is chosen. Motivated by these findings, they suggest adjustments to the relative positional encoding that helps with length generalization for addition.

**Strengths:**

The paper's perspective on length generalization is interesting. The authors look at different theoretical models to show how solutions to the LDHD problem can change as the models are refined. Although the empirical examples are limited, they have brief results on the empirical implications of their theoretical analysis.

**Weaknesses:**

I found the paper a bit hard to parse and making connections between different parts of the results was challenging to me. I believe if the authors use some of the remaining space within the page limit to discuss the high-level/intuitive connections of their results in more detail or discuss proof sketches, it would improve the paper's clarity. As it stands, I find it difficult to fully grasp the results.

I will list some of the points I had trouble following as questions below. It’s possible that I missed some aspects of the paper. Thus, I'm open to reconsidering my score if needed.

**Questions:**

1.  **Thm 1 & Cor 1**:
    - Can you explain why the min-degree interpolator cannot capture functions with $I(f)\not\subset [N_0]$?
    - Does Cor. hold for any projection $V$? Then, 1) what's the purpose of adding the projection to the RFM? and 2) how can designing $V$ (line 287) enable LDHD generalization if the target function has a dependence on higher dimensions?
2.  **Sec 4.2**: I don't understand the intuition behind the PLAA model:
    - on notations: 1) i didn't find $\mathcal{U}_N$ in notations. Is it referring to upper triangular matrices? 2) Is $x$ a sequence of $\{\pm1\}$? Then what does it mean "value of $x_i$ is identical to itself" (line 307)?
    - What is the motivation for defining PLAA this way? One thing that confuses me is that if the input is a sequence of all 1s, then PLAA always outputs zero by definition.
    - In Defn 6, the only difference to PLAA, is adding an upper triangular mask on top of the weights. while in PLAA, the learnable parameters are already constrained to be upper triangular. Where is the APE here?

3.  Can you also explain your proposed RPE-squared positional encoding? How do your previous results explain the improvement in the performance? Do you characterize how this PE changes the inductive bias which leads to generalization or is this just a heuristic suggestion?

---

> ### Author Response · Authors · 2024-11-20
>
> Thank you for your questions and valuable feedback.
>
> The core of our paper is the LDHD generalization perspective for length generalization and all our results revolve around this perspective. Motivated by the lack of a proper framework for general length generalization, we introduce the LDHD generalization perspective in Section 3. We show LDHD generalization is inherently difficult due to the No-Free-Lunch Theorem of LDHD generalization, and thus prior knowledge is required to achieve this generalization. Since prior knowledge is necessary for LDHD generalization, in Section 4, we analyze how different models encode different prior knowledge as inductive biases, which decide when a model can or cannot achieve LDHD generalization. In Section 5, we further connect the LDHD generalization perspective to two practical techniques, i.e., CoT and PE. We will include more explanations about the LDHD generalization perspective for length generalization and the results in Section 4 and the implications in Section 5 accordingly. Here are our responses to the listed questions.
>
> 1. **Thm 1 & Cor 1**
> - *Can you explain why the min-degree interpolator cannot capture functions with $I(f)\not\subset [N_0]$?*
>
> This is because a min-degree interpolator $f$ on $\mathcal{X}_{N_0}$ always satisfies $I(f) \subset [N_0]$.
>
> Otherwise, assume that $f$ is a min-degree interpolator on $\mathcal{X}_{N_0}$ but $I(f)\not\subset [N_0]$.
>
> Let $\tilde{f}(x)$ be the function constructed by fixing all $x_i$, $i\not\in [N_0]$ to $1$. By the construction, we have $\tilde{f}(x)=f(x)$ for all $x$ in $\mathcal{X}_{N_0}$, which is thus also an interpolator.
>
> However, $\text{DegP}(\tilde{f}) < \text{DegP}(f)$, contradicting the assumption that $f$ is the min-degree interpolator. Therefore, for a target function $f^*$ with $I(f^*)\not\subset [N_0]$, the min-degree interpolator on $\mathcal{X}_{N_0}$ cannot be equal to $f^*$.
>
> - *Does Cor. hold for any projection $V$? Then, 1) what's the purpose of adding the projection to the RFM? and 2) how can designing (line 287) enable LDHD generalization if the target function has a dependence on higher dimensions?*
>
> We require the columns of $V$, i.e., $v_1, \dots, v_r$, to be orthogonal to each other such that $\mathcal{V}$ is a linearly independent set. (To normalize, we suppose $V^\intercal V = I_r$ in the manuscript). Corollary 1 is implied by setting $V=I$ in Theorem 2.
>
> 1) We aim to illustrate two points of LDHD generalization via the random feature model with projection (RFMP):
>
>  - LDHD generalization is hard to achieve without good prior knowledge;
>  - Prior knowledge can be encoded by model design.
>
> To elaborate, on the one hand, Theorem 2 implies one projection choice can only achieve LDHD for a very restricted set of problems. If we do not know the right projection, the model may converge to a min-degree interpolator w.r.t. to a different linearly independent set and not be equal to the target function. On the other hand, if we have a good knowledge of the target function, we can simply encode this prior knowledge by choosing a proper projection in the RFMP.
>
> We choose RFM and its variant RFM because RFM is a classic model in the theoretical analysis. We think it beneficial to illustrate our points with such a classic model.
>
> 2) By choosing a different projection, the model can converge to a min-degree interpolator w.r.t. a different linearly independent set (denoted by $f$) such that $I(f)$ is not necessary a subset of $[N_0]$. This makes it possible to achieve LDHD generalization for the target function has a dependence on higher dimensions. If we have sufficient prior knowledge of the target function and design a $V$ such that the target function is of minimal degree profile w.r.t. $\mathcal{B}(V)$, then the RFMP with $V$ trained on $X_{N_0}$ is identical to the target function on $X_N$, achieving LDHD generalization.
>
> For example, consider the target function $f(x)=4 x_1 + 3 x_2$ and $N_0=1, N=2$. The min-degree interpolator on $X_{N_0}$ is $f_1(x)=4 x_1\neq f(x)$ does not achieve LDHD generalization. When we add projection and choose $V=[[0.8, 0.6]^\intercal [0.6, -0.8]^\intercal]$, we have $$B(V) = \\\{1, 0.8 x_1 + 0.6 x_2, 0.6 x_1 - 0.8 x_2, (0.8 x_1 + 0.6 x_2)(0.6 x_1 - 0.8 x_2)\\\}.$$
> The min-degree interpolator w.r.t. $B(V)$ is $f_2(x)=4 x_1 + 3 x_2=f(x)$, achieving LDHD generalization for the target function $f$ that has a dependence on the higher dimension than the training space $X_{N_0}$.

---

> > ### Author Response · Authors · 2024-11-20
> >
> > 2. **Sec 4.2:** *I don't understand the intuition behind the PLAA model:*
> > - *on notations: 1) i didn't find $\mathcal{U}_N$ in notations. Is it referring to upper triangular matrices? 2) Is $x$ a sequence of $\pm 1$? Then what does it mean "value of $x_i$ is identical to itself" (line 307)?*
> >
> > 1) Yes, $\mathcal{U}_N$ denotes the set of $N\times N$ upper triangular matrices. Thank you for pointing out this and we will add the missing definition.
> >
> > 2) This means we always choose the value matrix in the attention as $I$ and thus the value of $x_i$ $W_V x_i = x_i$. This is a simplification such that we can focus on the impact of the position embeddings on the attention score.
> >
> > - *What is the motivation for defining PLAA this way? One thing that confuses me is that if the input is a sequence of all 1s, then PLAA always outputs zero by definition.*
> >
> > We define PLAA in this way for a simplified model that abstracts the effect of position embedding on the LDHD generalization of decoder-only Transformers. We'd like to further elaborate on how we define this model step by step.
> >
> > Step 1. The linear attention at query $n$ can be denoted by $f_{\text{LA}}(x;W_Q, W_K, W_V, B)=\sum_{i\leq n} ((W_Q x_n)^\intercal W_K x_i + B_{i,n}) W_V x_i$, where $B$ is the position bias.
> >
> > Step 2. To focus on the impact of the position embeddings, we fix $W_V=I$ and make the attention score position-only. Then we have $$f_{\text{PLA}}(x; B)= \sum_{i\leq n} B_{i,n} x_i = \sum_{i\leq n} e_i^\intercal B e^n x_i.$$
> >
> > Step 3. Now the attention is position-only. We rewrite $B$ as $A$ and restrict $A$ to an upper triangular matrix. Then we have $f_{\text{PLA}}(x; A)=x^\intercal A e_n$.
> >
> > Step 4. In the prediction of the decoder-only Transformer, the query position $n$ provides extra information about the scale of the instance. Therefore, in the LDHD generalization, we assume the dimension of each instance is known by the model, which is reflected by the query position.
> >
> > To capture the intuition, we introduce the notation $n(x)=\text{argmax}_i \\\{x_i=-1\\\}$, i.e., the last dimension that is $-1$. In the LDHD generalization, $n(x)$ is the lowest dimension of the subspaces containing $x$ and thus reflects the dimension of $x$. We take $n(x)$ as the query position of $x$.
> >
> > For example, if $x=[1, 1, 1, -1, 1]^\intercal$, we can consider $x$ as $4$-dimensional and $n(x)=4$ is the query position. In other words, the query position $n(x)$ "advices" the dimension of $x$. Hence, we obtain $f_{\text{PLAA}}(x;A)=x^\intercal A e_{n(x)}$, i.e., the PLAA model in Definition 5.
> >
> > If the input $x$ is a sequence of all 1s, then $x$ is "empty" because we only consider the dimensions of the last $-1$ or lower as informative. Therefore, we fix the output of the all-one sequence to be consistent with the definition of PLAA.
> >
> > - *In Defn 6, the only difference to PLAA, is adding an upper triangular mask on top of the weights. while in PLAA, the learnable parameters are already constrained to be upper triangular. Where is the APE here?*
> >
> > The key difference of Defn 6 is that it reparameterizes the attention matrix $A$ with matrix factorization, i.e., $P^\intercal P$. To emphasize this reparameterization, we use the upper triangular mask to denote the upper triangular constraint, writing $A$ as $M_N^u \circ P^\intercal P$. The learnable parameters become $P$ instead of $A$. This reparameterization leads to a different inductive bias (Theorem 4) from that of the direct parameterization using $A$ as the learnable parameters directly (Theorem 3).
> >
> > We then explain why this reparameterization corresponds to APE. In APE, we typically add an APE $p_i$, which is a learnable embedding, with a token embedding $x_i$. For the query $n$ and the key at $i$, we have $(x_n + p_n)^\intercal W_Q^\intercal W_K (x_i + p_i)$. If we consider position-only attention, this becomes $p_n^\intercal W_Q^\intercal W_K p_i = e_n^\intercal P^\intercal W_Q^\intercal W_K P e_i$. Without loss of generality, we "absorb" $W_Q$, $W_K$ into the learnable parameters $P$, resulting in $e_n^\intercal P^\intercal P e_i$. Combining the upper triangular constraint, the attention matrix becomes $M_N^u \circ P^\intercal P$. Therefore, a PLAA model with APE can be denoted by
> > $$
> >     f_{\text{PLAA}}^{\text{APE}}(x;P) = x^\intercal \left(M_N^u \circ P^\intercal P\right) e_{n(x)}.
> > $$

---

> > > ### Author Response · Authors · 2024-11-20
> > >
> > > 3.
> > > - *Can you also explain your proposed RPE-squared positional encoding?*
> > >
> > > The proposed RPE-square illustrates how the principle that one needs to consider both the inherent LDHD generalization in the latent space and the data format in position embedding design.
> > >
> > > We then show how RPE-Square is designed according to the principle.
> > > - LDHD generalization in the latent space. For the addition (with CoT), the LDHD generalization in the latent space can be addressed by RPE. Note that the "position" in the latent space is the dimension.
> > > - The data format mapping of URF. The URF can be described as follows: concatenate the first elements of the dimensions (up to the "highest" nonzero element), a "+", the second elements of the dimensions (up to the "highest" nonzero element), a "=", and finally the third elements of the dimensions (up to the "highest" non-"*" element). To handle the data format, we consider the relations between a latent variable and its URF string. We find that the "position" of an element in the latent variable can be identified by its relative distances to "[BOS]", "+", and "=".
> > >
> > > For example, consider the URF addition "3 2 1 + 6 5 = 9 7 1" and $N=5$. When predicting "7", the latent variable is $\left[(3, 6, 9), (2, 5, *),(1, 0, *),(0, 0, *),(0, 0, *)\right]$. The relative distances to "[BOS]", "+", and "=" are $(6, 2, -1)$, and thus "5" is the second element of the second dimension; the triplet of "9" is $(8, 4, 1)$ and thus "9" is the third element of the first dimension. To predict $k$-th digit, corresponding to $(n_{>0}, n_{>0}, k)$, we need to identiy the $(k-1)$-th digit of the first addend, the $k$-th digit of the first addend, the $(k-1)$-th digit of the second addend, the $k$-th digit of the second addend, and the $(k-1)$-th digit of the sum, corresponding to $(k-1,n_{<0},n_{<0})$, $(k,n_{<0},n_{<0})$, $(n_{>0},k-1,n_{<0})$, $(n_{>0},k,n_{<0})$, and $(n_{>0}, n_{>0}, k-1)$ (neglecting the boundary cases).
> > >
> > > The prior knowledge of the data format is that we need to compute the relative distances to some tokens. Combining RPE and the approach to deal with the data format, we obtain RPE-Square. The position embedding for the query $j$ and the key $i$ is determined by **the relative distance** between **the relative distance of $j$ to the position of some token $x_l$**, and **the relative distance of $i$ to the position of some token $k$**, parameterized by $R_{(j-l)-(i-k)}$. (The name RPE-Square comes from its "relative distance of relative distances" design.) Furthermore, we want to learn the special tokens (i.e., "[BOS]", "+", and "=" in URF) automatically. For this purpose, we adopt a "soft" design, taking a weighted average over all $l, k, 1 \leq l\leq j, 1\leq k\leq i$. The weight of $R_{(j-l)-(i-k)}$ is determined by the tokens and we use an attention-style design, i.e., the multiplication between the attention score of the query $x_j$ and the key $x_l$ and the attention score of the query $x_i$ and the key $x_k$.
> > >
> > > - *How do your previous results explain the improvement in the performance?*
> > >
> > > The results in Theorem 5 and Corollary 3 imply that RPE introduces the correct inductive bias leading to the LDHD generalization in the latent space for the addition (with CoT).
> > >
> > > The LDHD generalization framework implies that it is necessary to consider both the inherent LDHD generalization in the latent space and the data format in the position embedding design to achieve length generalization. While both RPE and RPE-Square can handle the LDHD generalization in the latent space, RPE-Square further deals with the URF data format. This explains why RPE-Square has superior performance to RPE in the length generalization of the URF addition.

---

> > > > ### Author Response · Authors · 2024-11-20
> > > >
> > > > - *Do you characterize how this PE changes the inductive bias which leads to generalization or is this just a heuristic suggestion?*
> > > >
> > > > Our work does not provide a strict theoretical characterization of how RPE-Square changes the inductive bias in the practical Transformer, which is very difficult due to the complexity of the practical model. However, the LDHD framework offers a structured perspective such that RPE-Squre goes beyond being a mere heuristic suggestion.
> > > >
> > > > The LDHD generalization framework emphasizes the need to address both the inherent LDHD generalization in the latent space and the data format.
> > > >  1) For the inherent LDHD generalization in the latent space, we theoretically show how different PEs lead to different inductive biases in Section 4.1 in the simplified model PLAA. Corollary 3 specifically implies why RPE helps LDHD generalization for the addition with CoT. This insight suggests that an effective PE for length generalization in the URF addition task should be built upon RPE.
> > > >  2) For the data format, we show above that we need to consider the distances to some special tokens to handle the URF data format. This insight directly informs the design of RPE-Square, which incorporates the "relative distance of relative distances" concept to align with the data format.
> > > >
> > > > In summary, the design of RPE-Square is a natural implication of the LDHD generalization framework and not a heuristic developed without foundation. Our work provides a structured explanation for its effectiveness and highlights its grounding in the length generalization principle.

---

> > > > > ### Comment · Reviewer_w7k4 · 2024-11-24
> > > > >
> > > > > Thank you for the clarifications.
> > > > >
> > > > > The responses help clarify the scope and message of the paper. Still, I find the current format of the paper difficult to follow. Incorporating the examples, intuitions, and insights provided in the rebuttal would greatly improve the paper and strengthen the connections and flow between sections.
> > > > >
> > > > > I am willing to raise my score to 6 if these changes are incorporated into the final manuscript.
> > > > >
> > > > > One remaining question:
> > > > >
> > > > > You state, *"The results in Theorem 5 and Corollary 3 imply that RPE introduces the correct inductive bias..."* Does Theorem 5 directly address the addition task? I assume it does not, as the theory focuses on Boolean functions. The implication seems to be more along the lines of observing a change in inductive bias through RPE in Boolean tasks, which could extend to other setups. If that’s the case, the claim might overstate the connection, and it would be better to phrase it more cautiously.

---

> > > > > > ### Author Response · Authors · 2024-11-25
> > > > > >
> > > > > > Thank you for your comment and for raising your score. We have uploaded a revised manuscript that incorporates the examples, intuitions, and explanations in the rebuttal as suggested. To make our revisions easy to identify, all major updates in the manuscript have been highlighted in blue text.
> > > > > >
> > > > > >
> > > > > > **Response to the remaining question**
> > > > > >
> > > > > > Thank you for pointing out the potential overstatement of the connection. While Theorem 5 and Corollary 3 directly focus on Boolean functions, its implications extend to tasks like addition through a transformation to Boolean domains. Specifically, the addition task, defined on a finite domain, can be represented in binary form, allowing the problem to be analyzed using Boolean functions. This connection supports the intuition that RPE introduces an inductive bias beneficial to the length generalization of such tasks.

---

### Official Review · Reviewer_HTK8 · 2024-11-05

**Soundness:** 3
**Presentation:** 2
**Contribution:** 2
**Rating:** 6
**Confidence:** 3

**Summary:**

By studying length generalization through the lens of generalization from lower to higher dimensional subspaces, the paper provides results that explain why this length generalization is hard to achieve with modern architectures.

More specifically, the paper tackles the problem of length generalization in modern deep learning architectures. A representative example is the addition of numbers that are larger than any numbers that have appeared in the training data. The paper's main approach is to frame this kind of generalization as a generalization ability to embeddings that come from a subspace of larger dimension than the dimension of the training data and is termed Low-Dimension-to-High-Dimension (LDHD) generalization.

Theorem 1 establishes a no free lunch theorem stating that, without introducing additional assumptions (e.g., by having priors on the function that is being learned by incorporating some kind of inductive bias), it's not possible to generalize to a higher dimensional subspace. Intuitively, this is because the additional dimensions appearing in the test set are orthogonal to those in the training set and therefore it's by construction impossible to infer anything about them from the training data. The paper then looks at different models and studies their structural biases.

Theorem 2 establishes that, in the limit, the random feature model with projection converges to a minimum degree interpolator with respect to an independent set defined by the projection. This  implies that LDHD generalization is only achievable for a restricted set of functions that depend on the subspace of the training set. A similar result is shown for a certain class of decoder only transformers. An additional result is provided for transformers with absolute positional encodings, where it is shown that if the rank of the attention matrix required for the target function exceeds the dimension of the subspace that the training data live in, then LDHD generalization is not possible. Finally, a result is given for generalized relational positional encodings where again we are shown that in the limit, training converges to a minimum degree interpolator with respect to the independent sets induced by the positional encodings. These results all reinforce the idea that generalizing to arbitrary length sequences with some standard ML models is going to be challenging.
These ideas are used to argue that Chain of Thought is potentially effective in length generalization partly because it doesn't increase the dimensionality of the space for test data.

Finally, some positional encodings are proposed to alleviate the challenges that arise from using an URF encoding.

**Strengths:**

- This subspace-centric perspective on length generalization is interesting.
- The results on decoder-only transformers build nicely on previous work and provide a nice connection to the rank of the attention.
- rpe squared encodings seem to work for URF.

**Weaknesses:**

I find confusing the way certain sections of this paper are written which makes the overall story and contribution hard to tease out:

- What is the motivation behind the random model with projections? The result about the Fourier basis has been established in previous work. The paper claims around line 262 that not all RFMP models converge to the min-degree interpolator. However, theorem 2 esentially establishes that if we do a change of basis for the data then the min-degree interpolator is achieved on the projected fourier basis.
I don't see why theorem 2 is necessary to state corollary 1. Doesn't corollary 1 follow from the theorem in Abbe et al's paper?
I think it's unclear what exactly theorem 2 contributes to the discussion since it follows straightforwardly from the original result.

- Since chain of thought appears to improve results in those kinds of tasks, it seems important to understand how it fits into the LDHD generalization perspective. The paper provides a short discussion suggesting that CoT avoids increasing the dimensionality of the hidden space in which the data lie on and says that it 'extends the field' that is used by allowing for a symbol with undetermined values. It is not explained why this could be helpful (though it is suggested in the paper) to learn the correct target concept nor what quantitative predictions one can extract from this framing of CoT.  Why isn't CoT enough to deal with the generalization issue?

- I don't see how the proposed positional encodings connect to the contributions of this paper and the analysis that precedes them (line 431 claims that's the case). The subspace-centric view as I understand it is introduced so that the generalization problem is characterized in the hidden space. On the other hand, the paper seems to suggest that URF format makes generalization harder, or no? How does that connect with the comments about CoT being able to deal with the dimensionality of the hidden space? Why is CoT not enough and why are rpe-squared encodings needed? Do they somehow address the rank deficiency that is discussed in Corollary 2?

- The paper states in line 471: The design of RPE-Square illustrates how the principle can lead to position embeddings for better length generalization. What principle?  It addressed the URF formatting issue, but that seems orthogonal to the previous discussion.

Overall, I find the contribution of the paper potentially interesting, but it is quite hard to disentangle what the actual takeaways are from the paper. This is exacerbated by the writing being confusing and/or unclear in certain places that I have indicated. Until some of those things are cleared up I cannot recommend acceptance but I am obviously open to changing my score after the rebuttal.

**Questions:**

See above

---

> ### Author Response · Authors · 2024-11-20
>
> Thank you for your insightful feedback.
>
> 1. *What is the motivation behind the random model with projections?... I don't see why theorem 2 is necessary to state corollary 1. Doesn't corollary 1 follow from the theorem in Abbe et al's paper? I think it's unclear what exactly theorem 2 contributes to the discussion since it follows straightforwardly from the original result.*
>
> The motivation to consider the random model with projections includes two aspects:
> 1) LDHD generalization is hard to achieve without good prior knowledge;
> 2) Prior knowledge can be encoded by model design.
> We want to illustrate these two points with the classic random feature model and its variant, i.e., the random feature model with projection.
>
> Intuitively, in Theorem 2, we show that the model will always converge to the "simplest" solution w.r.t. the given set $\mathcal{B}(V)$ of "projected" functions. We need to know under which set the target function is the "simplest" otherwise LDHD generalization cannot be achieved.
>
> For example, consider the following two function: $f_1(x) = 4 x_1$ and
> $f_2(x) = 4 x_1 + 3 x_2$. We consider an LDHD generalization scenario where the training data are restricted to \{\pm 1\}\times\{0\} but the testing data can be sampled from \{\pm 1\}^2.
> Note that the training data for the two functions are the same but the training data cannot distinguish the two functions. Therefore, we have to rely on the prior knowledge, i.e., the choice of $V$ in the model design to achieve length generalization.
>
> For $V_1=I$, $B(V_1)=\\\{1, x_1, x_2, x_1 x_2\\\}$ and the learned model is $f_3(x) = 4 x_1 = f_1(x)$.
>
> For $V_2=[[0.8, 0.6]^\intercal\\ [0.6, -0.8]^\intercal]$, $B(V_2) = \\\{1, 0.8 x_1 + 0.6 x_2, 0.6 x_1 - 0.8 x_2, (0.8 x_1 + 0.6 x_2)(0.6 x_1 - 0.8 x_2)\\\}$ and the learned model is $f_4(x)= 4 x_1 + 3 x_2 = f_2(x)$.
>
> Neither of the designs can achieve LDHD generalization for $f_1(x)$ and $f_2(x)$ simultaneously. On the one hand, this reflects that LDHD generalization is challenging without sufficient prior knowledge of the target; on the other hand, the random model with projections demonstrates how different prior knowledge can be encoded via model design.
>
> Theorem 2 can be seen as a generalization of the Theorem in Abbe et al's paper.
> While Corollary 1 aims to demonstrate the challenge of LDHD generalization with a concrete example, we understand your concern that it may not fully show the contribution of Theorem 2 because it can be derived from the Theorem in Abbe et al's paper. We'd like to add another example (like the above one), which cannot be implied by Abbe et al's Theorem, to show the role of Theorem 2.

---

> > ### Author Response · Authors · 2024-11-20
> >
> > 2. *Since chain of thought appears to improve results in those kinds of tasks, it seems important to understand how it fits into the LDHD generalization perspective... It is not explained why this could be helpful (though it is suggested in the paper) to learn the correct target concept nor what quantitative predictions one can extract from this framing of CoT. Why isn't CoT enough to deal with the generalization issue?*
> >
> >
> > While CoT has been shown an effective technique to improve length generalization, previous works typically explain the success from the perspective of expressiveness, i.e., CoT increases the model's expressive power. However, this explanation is insufficient. Length generalization is not only a problem of expressiveness but also, more importantly, a problem of learnability. As our results from the LDHD generalization perspective imply, a model can fail to achieve length generalization even if it is sufficiently expressive. Additionally, the success of CoT cannot be explained if we define length generalization directly on the string length.
> >
> > LDHD generalization provides a new perspective to consider the role of CoT in length generalization. In the LDHD generalization framework, each instance is generated from a latent variable via a data format mapping. Considering the latent spaces without and with CoT, we find CoT may not be a simple data format change but lead to a different latent space. We may have prior knowledge of the LDHD generalization problem in the new latent space and can achieve this with a simple modification of the model (e.g., RPE).
> >
> > For example, in the addition problem, we show in Section 5.1 that the direct prediction of the sum without CoT and the digit-by-digit prediction, from lower digits to higher digits, with CoT, correspond to different latent space structures and different target concepts. While it seems complex to describe the rule to compute the addition as a whole, it is much easier to describe the digit-by-digit computation. To compute the $k$-digit in the sum, we only need to know the $(k-1)$-digits and $k$-digits in the two addenda and the $(k-1)$-digit in the sum, corresponding to the $(k-1)$-dimension and $k$-dimension of the latent variable. The relative distance between $(k-1)$ and $k$ is invariant to the scale of the problem with CoT. Therefore, in the latent space, our prior knowledge is to base our prediction on some fixed relative distances. This prior knowledge can be easily encoded by RPE.

---

> > > ### Author Response · Authors · 2024-11-20
> > >
> > > 3. *How does that connect with the comments about CoT being able to deal with the dimensionality of the hidden space? Why is CoT not enough and why are rpe-squared encodings needed? Do they somehow address the rank deficiency that is discussed in Corollary 2?*
> > >
> > >
> > > - *I don't see how the proposed positional encodings connect to the contributions of this paper and the analysis that precedes them (line 431 claims that's the case)...*
> > >
> > > Positional encodings are found closely related to the length generalization of Transformers in practice [1] [2] [3]. In Section 5.2, we explain why certain positional encodings can succeed in one scenario but may fail in even a slightly different one from the perspective of the LDHD generalization framework.
> > >
> > > More specifically, the LDHD generalization framework for length generalization suggests it is necessary to consider two aspects: the LDHD generalization in the latent space and the data format. A positional encoding needs to handle both aspects to achieve length generalization. In the addition (with CoT) example, RPE can address the LDHD generalization in the latent space. While RPE can deal with ARF at the same time, it cannot tackle URF.
> > >
> > > [1] Shaw, Peter, Jakob Uszkoreit, and Ashish Vaswani. "Self-attention with relative position representations." arXiv preprint arXiv:1803.02155 (2018).
> > >
> > > [2] Kazemnejad, Amirhossein, Inkit Padhi, Karthikeyan Natesan Ramamurthy, Payel Das, and Siva Reddy. "The impact of positional encoding on length generalization in transformers." NeurIPS (2024).
> > >
> > > [3] He, Zhenyu, Guhao Feng, Shengjie Luo, Kai Yang, Liwei Wang, Jingjing Xu, Zhi Zhang, Hongxia Yang, and Di He. "Two Stones Hit One Bird: Bilevel Positional Encoding for Better Length Extrapolation." ICML (2024).
> > >
> > > - *On the other hand, the paper seems to suggest that URF format makes generalization harder, or no?*
> > >
> > > URF format cannot be handled by RPE and is more "complex" than ARF. Therefore, we need a dedicated design to cope with the URF. In this sense, the length generalization of the URF addition is harder.
> > >
> > > - *How does that connect with the comments about CoT being able to deal with the dimensionality of the hidden space? Why is CoT not enough and why are rpe-squared encodings needed?*
> > >
> > > CoT and URF reflect the two aspects of length generalization, i.e., the LDHD generalization in the latent space and the data format, respectively.
> > >
> > > CoT is for the LDHD generalization in latent space. As discussed above (Question 2), CoT leads to a latent space, in which the LDHD generalization is "simple" and can be handled by RPE, while it is much more different to deal with the LDHD generalization in the latent space of the addition without CoT.
> > >
> > > URF is a data format. The URF addition (with CoT) and the ARF addition (with CoT) have the same latent space but different formats. While RPE happens to tackle the ARF format simultaneously, it cannot handle the URF.
> > >
> > > CoT is not enough because it is for the LDHD generalization in latent space but not for the data format. The rpe-squared encoding is needed to remedy RPE to deal with the URF data format.
> > >
> > > - *Do they somehow address the rank deficiency that is discussed in Corollary 2?*
> > >
> > > CoT may address the rank deficiency potentially. While CoT does not change the low-rank bias of APE because the bias is determined by the model and the training algorithm, it can change the latent space, leading to a different target function. If we have a CoT strategy such that the target $A^*$ is the lowest rank in the new latent space, APE could achieve length generalization with this CoT strategy.
> > >
> > > RPE-Square can be seen to consist of RPE and the extra design to handle the URF format. (To avoid repetition, further elaboration of RPE-Square is delayed to our response to Question 4.) RPE addresses the rank deficiency by making the model converge to a different solution where the rank can be larger than $N_0$. The extra design does not address the rank deficiency. The extra design is for the URF data format, while the rank deficiency is a problem of LDHD generalization in the latent space.

---

> > > > ### Author Response · Authors · 2024-11-20
> > > >
> > > > 4. *The paper states in line 471: The design of RPE-Square illustrates how the principle can lead to position embeddings for better length generalization. What principle? It addressed the URF formatting issue, but that seems orthogonal to the previous discussion.*
> > > >
> > > >
> > > > The principle is to consider both the inherent LDHD generalization in the latent space and the data format in the position embedding design.
> > > > This principle is implied by the LDHD generalization perspective, showing this perspective can help to design a model with better length generalization performance.
> > > >
> > > > To further elaborate, we show how RPE-Square is designed according to the principle.
> > > > First, we consider the LDHD generalization in the latent space.
> > > > As discussed above, for the addition with CoT, the LDHD generalization in the latent space can be addressed by RPE.
> > > > Note that the "position" in the latent space is the dimension.
> > > > We then consider the data format mapping of URF. The URF first concatenates the first elements of the dimensions (up to the "highest" nonzero element), a "+", the second elements of the dimensions (up to the "highest" nonzero element),
> > > > a "=", and then the third elements of the dimensions (up to the "highest" non-"*" element). To handle the data format, we need to "recover" the latent variable from the string.
> > > > According to the URF data format mapping, we can identify the "position" of an element in the latent variable by its relative distances to "[BOS]", "+", and "=".
> > > >
> > > > We encode the prior knowledge of the data format that we need to compute the relative distances to some tokens.
> > > > Combining RPE and the approach to deal with the data format, we obtain RPE-Square. The position embedding for the query $j$ and the key $i$ is determined by **the relative distance** between **the relative distance of $j$ to the position of some token $x_l$**, and **the relative distance of $i$ to the position of some token $k$**, parameterized by $R_{(j-l)-(i-k)}$. (The name RPE-Square derives from its 'relative distance of relative distances' design.) Additionally, we soften the "hard" RPE-Square such that we can learn the tokens of interest automatically, resulting in the RPE-Square in the paper (lines 463-467).
> > > >
> > > > The following is a concrete example to further illustrate the intuition behind RPE-Square. Consider the URF addition "3 2 1 + 6 5 = 9 7 1" and $N=5$. When predicting "7", the latent variable is $\left\[(3, 6, 9), (2, 5, *),(1, 0, *),(0, 0, *),(0, 0, *)\right\]$. We can identify the "position" of an element in the latent variable given its relative distances to "[BOS]", "+", and "=". For instance, the relative distances to "[BOS]", "+", and "=" are $(6, 2, -1)$, and thus "5" is the second element of the second dimension. Similarly, the triplet of "9" is $(8, 4, 1)$, and thus "9" is the third element of the first dimension.
> > > >
> > > > To predict "7", i.e., the third element of the second dimension, corresponding to $(8, 5, 2)$, we need to identify the elements corresponding to $(1,n_{<0},n_{<0})$, $(2,n_{<0},n_{<0})$, $(n_{>0},1,n_{<0})$, $(n_{>0},2,n_{<0})$, and $(7, 4, 1)$, where $n_{<0}$ and $n_{>0}$ denote a negative number and a positive number, respectively. More generally, to predict $k$-digit, corresponding to $(n_{>0}, n_{>0}, k)$, we need to identiy the elements corresponding to $(k-1,n_{<0},n_{<0})$, $(k,n_{<0},n_{<0})$, $(n_{>0},k-1,n_{<0})$, $(n_{>0},k,n_{<0})$, and $(n_{>0}, n_{>0}, k-1)$ (we ignore boundary cases for simplicity).
> > > >
> > > > In summary, the main takeaway is: **the inherence of length generalization is better characterized by LDHD generalization in a latent space than the varying sequence length**. Centered around the LDHD generalization perspective, we show length generalization (LDHD generalization) is challenging and prior knowledge is necessary (Section 3), how different model designs can encode different prior knowledge as inductive biases leading to different LDHD generalization performance (Section 4), and how this perspective can help to understand or design better practical techniques (Section 5).
> > > >
> > > > We appreciate your questions and suggestions, which will help us improve the clarity of the manuscript.

---

> ### Comment · Reviewer_HTK8 · 2024-11-29
>
> I thank the authors for their detailed responses. After reading these carefully, I can't help but feel that those clarifications that have been provided need to be incorporated in the text. I noticed that other reviewers had similar issues with parsing the text and understanding the rationale/motivation behind some of the results. I believe the paper will benefit from some restructuring and from clearly organizing the main ideas that are needed to achieve length generalization.
>
> This along with stronger evaluations as some of the other reviewers have recommended will probably be sufficient to get this paper over the acceptance threshold. As it is, I find the contribution presented here potentially interesting but it's hard to ultimately figure out the significance of the results, partly because of the way they are written and explained. The updated manuscript is an improvement but in my opinion more work needs to be done, e.g., clearly explaining the obstacles originating from the format (URF) vs the challenge of subspace generalization and if/how those are related.

---

> ### Author Response · Authors · 2024-11-29
>
> Thank you for your additional comments and for taking the time to carefully review our rebuttal and updated manuscript. We greatly appreciate your constructive feedback and your acknowledgment of the improvements made so far.
>
> 1. *the significance of the results*
>
> We emphasize that the primary contribution of our work is the LDHD perspective for length generalization, which disentangles the inherent scaling generalization and the data format nuisance. Under the LDHD framework, we highlight varying string length is not a good characterization for length generalization and explain certain techniques can help with length generalization or not in different scenarios. We believe our results help to better understand the core of length generalization.
>
> 2. *the obstacles originating from the format (URF) vs the challenge of subspace generalization and if/how those are related.*
>
> The obstacles originating from the format (URF) and the challenge of subspace generalization are two independent aspects of the addition's length generalization.
>
> **The challenge of subspace generalization.** When we learn to solve the addition from only simple (small-scale) instances, the key is to learn the rule of how the next digit is determined from certain previous digits. Otherwise, we would not be able to solve more complex instances. This rule is solely determined by the type of the operator (i.e., the addition), remaining **invariant** w.r.t. the format in which the addition instances are presented. (In other words, if the rule changes, then it becomes another operator instead of addition.)
>
> The challenge of subspace generalization is to learn the rule of the *addition* from samples corresponding to low-dimension subspaces. This is highly non-trivial and requires prior knowledge to encode proper inductive bias. For the addition, "relative distance" introduces the right inductive bias **independent of the format**.
>
> For example, in the instance "4 5 6 + 7 8 9 = 1 4 6 1", when we are to predict "6" in the sum, we need to consider the digits "5" "6" in the first addend, the digits "8" "9" in the second addend, the digit "4" in the sum, and the result "6" is computed from these digits. We need to learn the rule about which digits are of interest and how these digits decide the result. Only by this rule can we generalize to more complex addition instances. This rule can be effectively represented in the subspace and is invariant to the addition format.
>
> **The obstacles originating from the format (URF)** Since the model processes a string rather than the latent variable, we cannot apply the rule characterized in the latent space directly but consider the correspondence between the latent variable and the input string, i.e., the data format mapping from latent space to the input space. The data format can be chosen **independent of the rule** defined in the latent space, i.e., the operator. For example, both ARF and URF are data formats that transform a latent variable representing an addition instance to a string, respectively. While the ARF and URF strings are different, they both correspond to the same operator, the addition.
>
> While the ARF mapping from the latent space to the string space is simple and can be handled by RPE simultaneously, the URF mapping is much more complex. The redundant zeros in the URF are omitted and the instances of the same scale can have different lengths, which cannot be simply captured by relative distance on the string. As a result, it requires extra design to cope with this obstacle originating from the format (URF).
>
> If there are any further concerns or specific points that require additional clarification, we would be more than happy to address them. We *respectfully* ask you to kindly consider revising your score if the remaining concerns are satisfactorily addressed.

---

> ### Comment · Reviewer_HTK8 · 2024-11-30
>
> OK, in my view those explanations need to be added in the text to clearly distinguish between the challenges originating from the format and those that come from LDHD subspace generalization. The impression I was given by reading the text was that LDHD generalization allows us to bypass any format considerations, but based on your explanation that isn't quite the case, since mappings like URF are non-trivial and they need additional care so that the model can handle them (e.g., specific positional encodings). This also relates to CoT since it can in some sense address LDHD but not the formatting issue.
>
> I will increase my score, but I don't think some of the details of the explanations provided in the rebuttal are communicated well in the actual paper.

---

> > ### Author Response · Authors · 2024-12-01
> >
> > Thank you for your thoughtful follow-up and for agreeing to increase your score. As the deadline (Nov 26) for manuscript updates has passed, we are unable to revise the text at this stage. However, we are fully committed to incorporating these clarifications in a future revision.

---

### Official Review · Reviewer_8Yei · 2024-11-08

**Soundness:** 2
**Presentation:** 2
**Contribution:** 3
**Rating:** 5
**Confidence:** 4

**Summary:**

This paper introduces Low-Dimension-to-High-Dimension (LDHD) generalization, a framework addressing the challenge of generalizing from low-dimensional training data to high-dimensional testing data, which is relevant in Out-of-Distribution (OOD) generalization scenarios. LDHD generalization captures the scaling issues in reasoning tasks, where models trained on small-scale data must generalize to larger, more complex cases. The authors theoretically demonstrate that LDHD generalization is unattainable without incorporating inductive biases, formalized through a version of the “No-Free-Lunch Theorem.”

The paper explores LDHD generalization specifically in Boolean functions and presents findings that different architectures under Stochastic Gradient Descent (SGD) can achieve LDHD generalization if the model’s inductive bias aligns with the target function. It introduces Random Feature Models with Projection (RFMP) and position-only linear attention with Advice (PLAA) models and examines how their biases support LDHD generalization.

In practical applications, the authors propose the RPE-Square position embedding as an enhancement of the standard Relative Position Embedding (RPE). RPE-Square is designed to improve length generalization in cases with unaligned data formats. Additionally, the paper provides insight into Chain-of-Thought (CoT) reasoning, suggesting that it extends the latent space without increasing dimensionality, thus supporting LDHD generalization.

The paper contributes a new perspective on generalization challenges in reasoning tasks. It emphasizes the role of inductive biases and offers practical methods for achieving length generalization in complex models.

**Strengths:**

Broader Empirical Validation:
While the theoretical analysis and specific experiments (e.g., RPE-Square’s impact) are well-supported. However, the paper would benefit from a broader empirical evaluation across different models and datasets, especially complex reasoning tasks beyond Boolean functions. The current examples, while illustrative, may not fully capture the diversity of real-world data distributions. Including results from larger-scale datasets, such as multi-step arithmetic tasks or complex symbolic logic problems, would more effectively validate LDHD generalization and highlight its practical applicability.
Additionally, comparisons with alternative approaches to length generalization (such as recent Transformer variants or other positional encoding methods) would provide a more comprehensive view of where LDHD generalization stands in practice.

Explicit Justification of Assumptions:
Certain assumptions, such as those in the matrix factorization regularization conjecture (Assumption 2), are crucial to the theoretical claims but have not been fully explored. While empirical results partially support these assumptions, more discussion of their generalizability and limitations would be helpful. For example, analyzing situations where Assumption 2 may not hold could offer insights into potential failure cases, enhancing the robustness of the claims.

Increased Clarity in Technical Sections:
Some theoretical sections, particularly regarding the degree-profile bias and proofs for the PLAA models, are dense and may be challenging for readers less familiar with these concepts. Adding more intuitive explanations alongside formal definitions could make these sections more accessible. Diagrams illustrating key ideas, such as the shift from low-dimensional to high-dimensional spaces or the impact of different inductive biases, would also aid comprehension.
Furthermore, clarifying how specific model components contribute to LDHD generalization in practical tasks (beyond theoretical guarantees) could make the theoretical claims more relatable and actionable for practitioners.

Deeper Analysis of LDHD Limitations:
The LDHD generalization framework is theoretically strong, but discussing limitations or boundary conditions more explicitly would be beneficial. For instance, scenarios where LDHD generalization may fail—even with the proposed inductive biases—or limitations in scaling to very high dimensions could be explored. This would provide a balanced perspective on the framework’s applicability.

More Contextualization within Prior Work:
While the paper acknowledges related literature, it could provide a deeper comparison with recent works on length generalization (e.g., Ahuja & Mansouri 2024, Zhou et al. 2024) and other inductive bias approaches in Transformer architectures. Explaining how LDHD generalization and RPE-Square offer improvements or distinct advantages over these approaches would strengthen the contextual foundation. Additionally, discussing whether existing models like Recursive Neural Networks (RNNs) or Graph Neural Networks (GNNs) could benefit from LDHD generalization principles would broaden the relevance of the contributions.

Position Embedding Design Evaluation:
The RPE-Square embedding demonstrates promise, but a more thorough ablation study evaluating its impact under various configurations (e.g., different distance metrics or special token arrangements) would provide a more detailed understanding of its robustness. Testing RPE-Square in domains beyond the addition task, such as symbolic reasoning or multi-step logical tasks, could further establish its versatility and performance consistency across data formats.

In summary, enhancing empirical validation, refining theoretical assumptions, and providing further clarity and contextualization would elevate the strength and reach of this paper’s contributions. These additions would also help bridge the gap between theoretical insights and practical applicability in diverse model architectures and reasoning tasks.

**Weaknesses:**

The paper introduces a valuable perspective with LDHD generalization; however, it could be strengthened by addressing certain weaknesses in both its theoretical and experimental approaches.

Limited Empirical Scope:
While the theoretical contributions are strong, the experimental validation is limited in scope. The paper primarily demonstrates LDHD generalization through RPE-Square on specific cases, such as addition tasks. Testing a broader range of tasks, such as multi-step logical reasoning, symbolic manipulation, or mathematical operations, would provide a more convincing demonstration of the applicability of LDHD generalization. Expanding beyond Boolean functions to real-world data sets in NLP or other reasoning tasks would further validate the practicality of the approach.

Additionally, comparing the performance of RPE-Square with other recent advances in length generalization, such as transformer variants designed for reasoning (e.g., Kazemnejad et al., 2024) or models trained on complex arithmetic tasks (e.g., Lee et al., 2023), would highlight RPE-Square’s strengths and limitations.

Assumption Dependence and Generality:
Certain key assumptions in the theoretical framework, like those in the minimal-nuclear-norm-solution conjecture (Assumption 2), may not hold universally, potentially limiting the framework’s generality. The assumptions play a significant role in supporting the theoretical findings, so the paper could benefit from a more detailed examination of their applicability and limitations. For instance, providing empirical tests or theoretical discussions on where these assumptions might fail would offer a more balanced understanding of LDHD generalization’s robustness.

Exploring whether alternative assumptions or regularization methods could yield similar results would help generalize the framework. This would make the theoretical claims more robust and adaptable to a wider range of models.

Theoretical Accessibility and Clarity:
The paper’s dense theoretical sections, especially those regarding the degree-profile bias and proofs for the PLAA model, may be challenging for readers without a strong background in this area. Adding intuitive explanations or visual aids alongside the formal proofs could significantly improve readability and accessibility. A step-by-step breakdown of key concepts, particularly for the degree profile and min-degree interpolator, would make these sections easier to understand.

Diagrams illustrating key ideas, such as low-to-high dimensional shifts or how different inductive biases support LDHD generalization, would also help clarify the framework’s mechanisms.

Deeper Analysis of LDHD Generalization’s Practical Implications:
The paper would benefit from a more explicit discussion of the practical limitations of LDHD generalization. While LDHD is theoretically promising, acknowledging potential challenges—such as cases where the high-dimensional space differs fundamentally from training distributions—would strengthen the work. Highlighting when LDHD generalization may fail, even with well-designed inductive biases, would provide a balanced perspective.
Furthermore, discussing the implications of the LDHD framework for models beyond transformers, such as Graph Neural Networks (GNNs) or Recursive Neural Networks (RNNs), would broaden its relevance and demonstrate the approach's flexibility.

Limited Contextualization within Recent Length Generalization Research:
Although the paper addresses related work, it could better contextualize LDHD generalization relative to recent advances. Comparing LDHD with other generalization frameworks, especially recent developments in length generalization (e.g., Ahuja & Mansouri, 2024; Zhou et al., 2024), would clarify its unique contributions and limitations.
Providing insights into whether the LDHD framework could be adapted for methods beyond transformers, such as RNNs or CNN-based architectures, would also enrich the discussion. Additionally, examining whether other position encoding methods, such as rotary embeddings, would benefit from an LDHD perspective could reveal additional practical insights.

Broader Evaluation of RPE-Square Design Choices:

While RPE-Square demonstrates improvements in the addition task, a more comprehensive evaluation of its design would be valuable. The current analysis could be strengthened by testing RPE-Square under various conditions, such as different distance metrics, transformations, or arrangements of special tokens. This would provide a deeper understanding of its robustness across task variations.

Extending RPE-Square’s evaluation to additional domains, such as symbolic logic, planning, or other reasoning-intensive tasks, would demonstrate its versatility and generalizability. Additionally, an ablation study to understand the specific impact of each component (e.g., relative vs. absolute position aspects) would clarify the primary drivers of its performance gains.

The paper would benefit from broader empirical validation, expanded theoretical clarity, and deeper contextualization. These enhancements would make LDHD generalization’s theoretical and practical implications more robust, accessible, and impactful, better aligning the work with its stated goals.

**Questions:**

Broader Task Validation for LDHD Generalization:
Question: Have you considered validating LDHD generalization on a wider set of tasks beyond Boolean functions and addition problems? For instance, tasks involving more complex arithmetic, symbolic reasoning, or planning tasks?
Suggestion: Expanding the evaluation to include a broader range of reasoning-intensive tasks would demonstrate the generality and practical applicability of LDHD generalization. Additionally, comparing your findings with those of other approaches to length generalization, such as Kazemnejad et al. (2024) and Lee et al. (2023), could add value by highlighting their relative strengths and limitations.

Justification and Generality of Assumptions:
Question: Can you provide more insights into the practical applicability of the minimal nuclear norm solution conjecture (Assumption 2) and other key assumptions? Are there scenarios in which these assumptions may not hold, and how might that affect LDHD generalization?
Suggestion: It would be helpful to clarify the scope of these assumptions, perhaps by discussing or empirically validating their generality across different architectures. Exploring whether alternative assumptions or regularization techniques could produce similar results would provide a more robust foundation for your theoretical contributions.

Improving Theoretical Accessibility:
Question: Would you consider adding intuitive explanations or diagrams alongside the formal theoretical sections to enhance accessibility, especially for concepts like the degree-profile bias and min-degree interpolators?
Suggestion: While the formal proofs are well-structured, additional explanations or illustrations could help a broader audience appreciate the theoretical underpinnings of LDHD generalization. Visual aids representing the concept of low-to-high dimensional shifts, for example, might clarify these ideas.

Implications of LDHD Generalization for Other Architectures:
Question: Do you envision LDHD generalization principles applying to architectures beyond transformers, such as Graph Neural Networks (GNNs) or Recursive Neural Networks (RNNs)? If so, could you provide insights on how LDHD generalization might be adapted to these structures?
Suggestion: Discussing the potential applicability of LDHD generalization principles to other architectures could broaden the relevance of your contributions. Exploring cases where GNNs or RNNs might also benefit from LDHD-inspired inductive biases would make the work more versatile and impactful.



Broader Analysis of RPE-Square Design Choices:
Question: Could you provide more details on the design decisions for RPE-Square, such as the choice of distance metrics or the arrangement of special tokens? Have you tested alternative configurations? If so, how do they affect performance?
Suggestion: Analyzing the impact of design choices within RPE-Square, possibly through an ablation study, would provide a deeper understanding of what makes this embedding effective. Additionally, extending RPE-Square evaluations to other domains (e.g., symbolic logic or planning tasks) would clarify its generalizability.
Explicit Discussion on Potential Limitations:
Question: Are there specific conditions or model architectures where LDHD generalization might fail, even with well-designed inductive biases? Could you outline any known limitations or failure modes?
Suggestion: A section explicitly discussing possible limitations or boundary conditions for LDHD generalization would provide a more balanced perspective on the approach’s applicability. This could include insights into cases where training and testing distributions diverge significantly or where high-dimensional testing distributions have features not represented in low-dimensional training data.
Relation to Recent Work in Length Generalization:
Question: How does LDHD generalization relate to recent studies on length generalization, such as Ahuja & Mansouri (2024) and Zhou et al. (2024)? Could you provide a deeper comparison of LDHD positions relative to these approaches?
Suggestion: Discussing how LDHD generalization contrasts with or complements other length generalization strategies would provide a clearer contextual foundation. This would also help highlight your framework’s unique contributions, especially regarding scaling challenges and inductive bias requirements.

Addressing these questions and suggestions would clarify the framework’s applicability, limitations, and comparative advantages, helping to deepen the impact and reach of the paper’s contributions.

---

> ### Author Response · Authors · 2024-11-20
>
> Thank you for your very detailed review and constructive suggestions.
>
> **Broader Task Validation for LDHD Generalization**
> *Question: Have you considered validating LDHD generalization on a wider set of tasks beyond Boolean functions and addition problems? For instance, tasks involving more complex arithmetic, symbolic reasoning, or planning tasks?*
>
>
> Since LDHD generalization is a general framework, it can be applied to the length generalization of various reasoning tasks.
>
> To show this, we additionally consider the length generalization in the copy task. In the copy task, given an input sequence $x_1 \dots x_n =$, the goal is to predict the copy of $x$. An instance is like "$x_1 \dots x_n = x_1 \dots x_n$.
> For length generalization, the training data only include instances whose $n \leq N_0$, while the model is evaluated on data with $N_0 < n \leq N$. In our experiment, we examine the scenario where $N_0=5$ and $N=10$. We compare the length generalization performance of the models with RPE and RPE-Square.
> The result is presented in the anonymous link: https://www.dropbox.com/scl/fi/6k2jdk1a4hecx55grypdy/length_generalization_copy.pdf?rlkey=y2dkdy16sxgqh882r05werbro&st=6j8w5kfb&dl=0. The results show that RPE-Square achieves near-perfect length generalization while RPE fails. The superiority of RPE-Square over RPE in the length generalization of the copy task demonstrates that LDHD generalization is valid beyond the addition task.
>
> More applications of LDHD generalization are left for future work.
>
> **Justification and Generality of Assumptions**
> *Question: Can you provide more insights into the practical applicability of the minimal nuclear norm solution conjecture (Assumption 2) and other key assumptions? Are there scenarios in which these assumptions may not hold, and how might that affect LDHD generalization?*
>
> The minimal nuclear norm solution conjecture depicts the inductive bias of the parameterization as matrix factorization, which tends to be low-rank. (Roughly, the matrix nuclear norm can be seen as a good reflection of the matrix rank. There are extensive technical discussions on the relations between the two quantities in the literature of low-rank optimization, which is beyond the scope of this work.) While theoretical conditions under which the assumption holds are technical, some works find this tendency exists in various practical scenarios.
>
> **Improving Theoretical Accessibility**
> *Question: Would you consider adding intuitive explanations or diagrams alongside the formal theoretical sections to enhance accessibility, especially for concepts like the degree-profile bias and min-degree interpolators?*
>
> We appreciate your advice and are happy to add intuitive explanations and diagrams. Here, we'd like to illustrate some of the most important concepts intuitively.
> - Low-Dimension-to-High-Dimension Shift: Low-dimension-to-high-dimension shift is a type of out-of-distribution shift with the special structure that the testing domain is higher-dimensional than the training domain. For an intuitive illustration, we compare the in-distribution generalization, a typical OOD generalization with distributional shift, and the LDHD generalization in Figure 1 of the anonymous link: https://www.dropbox.com/scl/fi/r8jstagj1dkk0reqeo7vr/comparison_generalization.pdf?rlkey=enzv67jum577bfgtmvgtnlx2j&st=9a4ro5lr&dl=0.
> - Degree Profile: Intuitively, a degree profile depicts the overall distribution of the coefficients of items with different degrees in the Fourier expansion of a Boolean function, which is a polynomial. Roughly, the degree profile reflects the complexity of the Boolean function.
> - Min-Degree Interpolator: A min-degree interpolator of a dataset is the interpolator of the minimal degree-profile. Intuitively, a min-degree interpolator is the "simplest" function that fits the data.

---

> > ### Author Response · Authors · 2024-11-20
> >
> > **Implications of LDHD Generalization for Other Architectures**
> > *Question: Do you envision LDHD generalization principles applying to architectures beyond transformers, such as Graph Neural Networks (GNNs) or Recursive Neural Networks (RNNs)? If so, could you provide insights on how LDHD generalization might be adapted to these structures?*
> >
> > Since the LDHD generalization considers the common scaling inherence of the length generalization problem, it is not restricted to Transformers. The main insight of the LDHD generalization principle is to disentangle the length (scale) generalization problem into two aspects: the LDHD generalization in the latent space and the data format. This insight can be applied to various architectures.
> >
> > For example, in the scale generalization problem of graphs, we can consider which reflects the "inherent" scale and which is the "format" of a graph. When we are to design a GNN to achieve scale generalization, we can consider the component to handle LDHD generalization in the latent space and the component to tackle the format, respectively, and then combine the two components. For the length generalization of RNNs, we can also deal with the LDHD generalization and the data format separately, adapting RNNs to handle the two aspects, respectively, and combining the two modifications.
> >
> >
> > **Broader Analysis of RPE-Square Design Choices**
> > *Question: Could you provide more details on the design decisions for RPE-Square, such as the choice of distance metrics or the arrangement of special tokens? Have you tested alternative configurations? If so, how do they affect performance?*
> >
> > The design of RPE-square is guided by the principle that one needs to consider both the inherent LDHD generalization in the latent space and the data format in position embedding design.
> > - The LDHD generalization of the addition (with CoT) in the latent space can be handled by RPE.
> > - The URF format can be handled by relative distances to "[BOS]", "+", and "=". The three relative distances reflect the correspondence between a latent variable and a string.
> > Combining the two aspects, we obtain the "relative distance of relative distances" design and parameterize the position embeddings by $R_{(j-l)-(i-k)}$ for the query at $j$, the key at $i$, and all $1\leq l\leq j$, $1\leq k\leq i$. Furthermore, we want to learn the special tokens (i.e., "[BOS]", "+", and "=" in URF) automatically and thus adopt a "soft" design, taking a weighted average over all $l, k, 1 \leq l\leq j, 1\leq k\leq i$. The weight of $R_{(j-l)-(i-k)}$ depends on the tokens, i.e., the multiplication between the attention score of the query $x_j$ and the key $x_l$ and the attention score of the query $x_i$ and the key $x_k$.
> >
> > Different configurations of RPE-Square may lead to different inductive biases and thus could affect the length generalization performance. We introduce RPE-Square as an example to show the power of the LDHD generalization perspective when designing models with better length generalization. While exploring how configurations of RPE-Square can influence its performance is an interesting direction, we leave it for future work.
> >
> > **Explicit Discussion on Potential Limitations**
> > *Question: Are there specific conditions or model architectures where LDHD generalization might fail, even with well-designed inductive biases? Could you outline any known limitations or failure modes?*
> >
> > One limitation of LDHD generalization is that it does not explain the scenario where the data format makes the length generalization simpler instead of more complex than the LDHD generalization. While LDHD generalization disentangles the latent LDHD generalization and the data format, these two aspects could have joint effects that promote length generalization.

---

> > > ### Author Response · Authors · 2024-11-20
> > >
> > > **Relation to Recent Work in Length Generalization**
> > > *Question: How does LDHD generalization relate to recent studies on length generalization, such as Ahuja & Mansouri (2024) and Zhou et al. (2024)? Could you provide a deeper comparison of LDHD positions relative to these approaches?*
> > >
> > > Most recent studies focus on the length generalization of specific tasks. While these works do propose effective techniques to improve the length generalization in certain scenarios, they rarely discuss the general length generalization and why or why not the techniques can be applied to a different setting, potentially leading to the fragility of length generalization performance [1].
> > > This motivates us to consider a general framework that captures the common "core" of length generalization and guides our model designs for more robust length generalization.
> > >
> > > Several works, including Ahuja & Mansouri (2024) [2], discuss the length generalization in a general setting. They directly treat length generalization as the generalization from a short sequence to a long sequence,
> > > which may not precisely characterize the problem scaling inherence of length generalization. [3] mention the challenge of length generalization may not necessarily be captured by the sequence length but they do not formalize this intuition. By defining the generalization on the dimension of the latent space instead of the sequence length, the LDHD generalization framework is more consistent with the intuition of length generalization in practice, capturing the learning challenge of length generalization more precisely.
> > > The LDHD generalization perspective takes a step towards a general framework to analyze length generalization.
> > >
> > > [1] Zhou, Yongchao, Uri Alon, Xinyun Chen, Xuezhi Wang, Rishabh Agarwal, and Denny Zhou. "Transformers can achieve length generalization but not robustly." arXiv preprint arXiv:2402.09371 (2024).
> > >
> > > [2] Ahuja, Kartik, and Amin Mansouri. "On provable length and compositional generalization." arXiv preprint arXiv:2402.04875 (2024).
> > >
> > > [3] Abbe, Emmanuel, Samy Bengio, Aryo Lotfi, and Kevin Rizk. "Generalization on the unseen, logic reasoning and degree curriculum." In Proceedings of the 40th International Conference on Machine Learning, pp. 31-60. 2023.

---

> > > > ### Author Response · Authors · 2024-12-01
> > > > **End of Discussion Period Approaching**
> > > >
> > > > Thank you for your valuable feedback on our manuscript. With the discussion period nearing its conclusion, we kindly ask for your thoughts on our responses to your review. If you have any additional comments or concerns, we would be happy to address them promptly.

---

### Meta-Review · Area_Chair_Trpu · 2024-12-20

**Metareview:**

This paper investigates the Low-Dimension-to-High-Dimension (LDHD) generalization, where the training data is assumed to live in a low-dimensional subspace of the high-dimensional testing space.
The paper presents a "hardness result" for Boolean functions, showing that LDHD generalization is generally unattainable unless some form of inductive bias is introduced based on prior information.

While the paper offers some interesting insights into LDHD generalization, **all reviewers** agreed that the presentation could be significantly improved to help readers better understand the contributions of this work.
The authors actively engaged with the reviewers during the rebuttal phase; however, they were unable to convince all reviewers that the paper is ready for publication in its current form.

As a result, I recommend rejecting the paper.

**Additional Comments On Reviewer Discussion:**

The reviewers raised the following major concerns:

- Presentation (raised by **all reviewers**): This issue was only **partially resolved** by the authors during the rebuttal phase.
- Other clarifications: These include the significance of the results (raised by Reviewer HTK8) and the connection to other results (raised by Reviewers 8Yei and eFtA), etc. While the majority of these concerns were, in my opinion, successfully addressed by the authors, it appears that the manuscript requires thorough revision to fully incorporate these comments.

I have considered all of the above points in making my final decision.

---

### Decision · Program_Chairs · 2025-01-22

Reject